# LLaMoCo: Instruction Tuning of Large Language Models for Optimization Code Generation

## Abstract

Recent research on optimization using large language models (LLMs) typically involves either iterative next-step solution seeking or directly prompting LLMs to generate critical optimization codes. However, these methods often suffer from low computational efficiency, high sensitivity to prompt design, and a lack of domain-specific knowledge. We introduce LLaMoCo, the first instruction-tuning framework designed to adapt LLMs for solving optimization problems in a code-to-code manner. LLaMoCo features a comprehensive instruction set that includes code-style problem descriptions as input prompts and robust optimization codes from expert optimizers as target outputs. We then develop a novel two-phase learning strategy with a contrastive learning-based warm-up to enhance convergence during instruction tuning. Extensive experiments demonstrate that a CodeGen (350M) model tuned by our LLaMoCo yields a powerful domain-specific model for generating expert-level optimizers, achieving superior performance compared to GPT-4 Turbo and other competitors on both synthetic and realistic problem sets. The trained model and the usage instructions are available online.

## 1 Introduction

Nowadays, Large Language Models (LLMs) are posing a profound impact on human society (Floridi and Chiriatti, 2020; Lund and Wang, 2023) through their remarkable natural language understanding and ability to solve complex tasks (Biswas, 2023a; Lund and Wang, 2023; Biswas, 2023b). Optimization is very related to LLMs. Optimization is everywhere yet can not be easily solved by everyone (AhmadiTeshnizi et al., 2023). For now, most optimization problems from scientific or industrial scenarios are solved by hand, requiring expert-level knowledge to 1) formulate the problem, 2) solve the problem with the desired optimizer. This in turn hinders the widespread of optimization techniques. This raises a key research question: *Can LLMs tackle challenging optimization problems that are difficult for humans to address?* This question drives the core of our study in this paper.

In the literature, several works have explored the possibilities of using LLMs to solve optimization problems. A common and straightforward way is to iteratively prompt LLMs to generate better solutions through a multi-turn conversation process (Yang et al., 2023; Guo et al., 2023a; Liu et al., 2023a), sometimes incorporating the concept of in-context learning. This solution-to-solution process involves prompting the LLMs with initial or current best-so-far solutions and iteratively requesting improved solutions. While showing certain effectiveness, they can have several limitations: 1) the scale of target optimization tasks (e.g., in terms of the number of decision variables, historical solutions and newly generated solutions) is constrained by the context window length of LLMs; 2) the iterative process typically involves hundreds rounds of conversations, consuming significant resources or API callings; and 3) due to LLMs' sensitivity to prompt design, it is nontrivial to provide consistent prompts that ensure ideal outputs.

An alternative way involves directly prompting LLMs to generate optimization programs, either through reusing existing optimization toolboxes (AhmadiTeshnizi et al., 2023) or combining multiple optimizers to create novel ones (Pluhacek et al., 2023). It can be more efficient than the solution-to-solution methods for two reasons: 1) only a few rounds of conversation are needed to generate codes; and 2) the prompts and generated codes do not include solution information, making it compatible

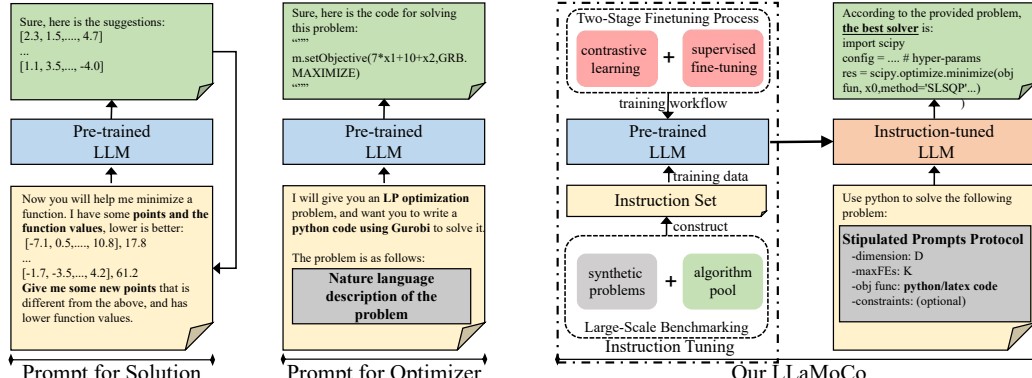

Figure 1: Conceptual overview of *LLMs for optimization*. **Left**: optimization through iteratively prompting for better solutions (e.g., (Yang et al., 2023)). **Middle**: optimization through prompting LLMs for optimizer configuration or code generation (e.g., (Romera-Paredes et al., 2024; Ahmadi-Teshnizi et al., 2023; Lange et al., 2024)). **Right**: our LLaMoCo framework, which is among the first to learn a domain-specific optimization model that generates expert-level optimization code (optimizers) from formatted prompts, tuned using a comprehensive instruction set.

with the problem scales. However, careful prompt crafting is crucial to ensure logical coherence. For example, OptiMUS (AhmadiTeshnizi et al., 2023) integrates hints about the target optimizer into the prompts, requiring a deep understanding of optimization domain knowledge. Additionally, LLMs pre-trained on a broad corpus often fail to generate customized optimizers tailored to a specific optimization problem instance due to the *lack of domain-specific expert knowledge* (Zhao et al., 2023), a limitation that extends to other complex tasks involving structured data, such as knowledge-base question answering and semantic parsing (Jiang et al., 2023; Xie et al., 2022).

In this paper, we propose **LLaMoCo**, a novel framework that adapts general-purpose **L**arge **La**nguage **M**odels for **o**ptimization **Co**de generation. Unlike methods based solely on prompt engineering, LLaMoCo learns a domain-specific model to generate expert-level optimizers tailored at the instance level. It leverages a well-formatted instruction set comprising code-to-code pairs of problem descriptions in Python or LaTeX and their corresponding high-performing optimizer implementations, refined through extensive benchmarking and hyper-parameter tuning. Once trained with our customized instruction set that contains diverse examples of effective optimizers for various problem instances with distinct landscapes, the model can generalize well to unseen optimization problems. To accelerate the training convergence, we introduce two-phase learning: first enhancing the latent representation through contrastive learning (Hadsell et al., 2006), followed by sequence-to-sequence instruction tuning with a designed loss function. LLaMoCo offers several unique advantages: 1) It generates optimization programs in a single round, making it more efficient for large-scale problems; 2) We standardize problem description prompts to structured codes, reducing prompt design effort and enhancing underlying pattern learning for superior optimization performance and zero-shot generalization; 3) LLaMoCo can generate more robust, expert-level optimizers compared to existing general LLMs. Figure 1 depicts the difference between our LLaMoCo and existing approaches.

Extensive experiments reveal the remarkably robust performance of our LLaMoCo, surpassing existing methods. Notably, we show that instruction tuning a small LLM like CodeGen-350M (Nijkamp et al., 2023) on domain-specific tasks significantly outperforms even very large models like GPT-4 (Achiam et al., 2023). Moreover, we also provide in-depth analyses of the two-phase adaptation strategy, training data sensitivity, and zero-shot generalization performance. **Our contributions are four folds**: 1) we introduce LLaMoCo, the first framework for adapting LLMs to generate expert-level optimizers; 2) we establish a code-to-code instruction set tailored for the optimization domain; 3) we propose a two-phase training strategy based on contrastive warm-up training; and 4) our LLaMoCo exhibits superior zero-shot generalization, efficiency, and robustness.

## 2 RELATED WORKS

**Fine-tuning LLMs.** Pre-trained LLMs can be refined by parameter updates on specific tasks through a fine-tuning process. We introduce two prominent fine-tuning strategies: Instruction

Tuning (IT) (Ouyang et al., 2022) and Alignment Tuning (AT) (Christiano et al., 2017; Ziegler et al., 2019), each serving distinct purposes. Generally, IT involves fine-tuning pre-trained LLMs using a moderate collection of formatted task instances (Wei et al., 2022a). The fine-tuning process typically includes two steps: 1) prepare instruction-formatted training examples by associating a task description with each task instance, which aids LLMs in understanding tasks (Sanh et al., 2022); and 2) leverage the prepared instruction set to fine-tune LLMs using a sequence-to-sequence supervised loss (Gupta et al., 2023). By incorporating a well-established task-specific instruction set, IT can effectively inject domain-specific knowledge into general LLMs. It enables the transfer of LLMs to specific experts in domains like medicine (Singhal et al., 2023), law (Huang et al., 2023) and finance (Zhang et al., 2023a). Differently, AT aims to correct unexpected behaviors of LLMs by aligning the models with human values and preferences (Ouyang et al., 2022; Ziegler et al., 2019). A practical algorithm for AT is the Reinforcement Learning from Human Feedback (RLHF) (Ziegler et al., 2019), which firstly estimates a reward model on a human-preference data collection via maximum likelihood. It then uses the learned reward model to provide feedback and post-trains the LLMs through Proximal Policy Optimization (PPO) (Schulman et al., 2017). A recent work named Direct Preference Optimization (DPO) (Rafailov et al., 2023) first reparameterizes the reward function based on the parameters of the pre-trained LLMs, saving the modelling and training of the reward function. DPO is mathematically equivalent to RLHF but is even more efficient, which is widely adopted in the latest LLMs such as Mistral 8x7B (Jiang et al., 2024). Our LLaMoCo is the first instruction-tuning framework for adapting general LLMs as an efficient and effective optimization tool, which addresses the unsatisfactory optimization performance due to the limited domain-specific knowledge of general LLMs.

**LLMs for code generation.** Generating code from natural language descriptions is exciting and complex (Zan et al., 2023; Chen et al., 2021). Although general-purpose LLMs such as GPT (Brown et al., 2020), Llama 2 (Touvron et al., 2023) and Mistral (Jiang et al., 2024) show competitive performance on the widely used LLM benchmarks including HumanEval (Chen et al., 2021), MBPP (Austin et al., 2021) and DS-1000 (Lai et al., 2023), their performance on a particular task may still be limited. Recent efforts have focused on developing LLMs specifically tailored for code generation. These models can be trained exclusively on code, such as AlphaCode (Li et al., 2022) and StarCoder (Li et al., 2023), fine-tuned from general LLMs, like Codex (Chen et al., 2021) and Code Llama (Roziere et al., 2023), or prompted from pre-trained LLMs such as FunSearch (Romera-Paredes et al., 2024). Notably, Codex shows that a 12B LLM can solve $72.31\%$ of complex programming tasks posed by humans. This success has led to the emergence of various Code LLMs, such as CodeGen (Nijkamp et al., 2023) that factorizes a potentially long specification into multiple steps to enhance program synthesis, and Code Llama that extends Llama 2 models through a cascade of fine-tuning steps. Other models such as Phi-2 (Javaheripi et al., 2023), InCoder (Fried et al., 2023) and CodeGeeX (Zheng et al., 2023) have also gained great attention. Nevertheless, the target scenario in our LLaMoCo is the optimization domain, where the optimization code generation is more challenging than normal code generation task due to the intricate semantic alignment issue and data imbalance issue. We in this paper propose a novel two-stage instruction tuning to address these challenges.

**LLMs as optimizers.** Optimization is crucial in numerous science and engineering fields but poses significant challenges. Unlike tasks such as language understanding, optimization problems are difficult for humans to solve without efficient algorithms, challenging LLM's reasoning and generalization abilities. Recent research has explored **prompting LLMs for solutions** of the given numerical optimization (Yang et al., 2023; Wang et al., 2024; Liu et al., 2023b), optimizer discovery (Pluhacek et al., 2023; Liu et al., 2024; Zhang et al., 2023b), or prompt optimization (Guo et al., 2023b) scenarios, which is based on prompt engineering and in-context learning (Min et al., 2022). Typically, these methods involve a set of candidate solutions for improvement, where LLMs receive prompts with these solutions and their objective values and then generate improved solutions iteratively until a termination condition is met. However, such a paradigm challenges the expertise of the general LLMs for optimization, which is less developed during their pre-training. To address this, the latest studies innovatively proposed prompting LLMs to behave like black-box optimizers, that is, instructing LLMs to perform mutation, crossover operations and elitism strategy (Liu et al., 2023a; Lehman et al., 2023; Chen et al., 2023; Brahmachary et al., 2024) on the candidate solutions. An eye-catching work of this line is EvoPrompting (Chen et al., 2023), which is surprisingly capable of finding neural network architectures with state-of-the-art performance. However, these approaches have limitations in efficiency due to the need for extensive iterations.

In contrast, several studies share a similar ambition with our LLaMoCo: **prompting LLMs directly for pieces of executable optimization programs**. An emerging research area is LLMs for Algorithm Design (Liu et al., 2024) in the combinatorial optimization domain, that is, applying LLMs at the algorithm design level to search and refine algorithm programs in an evolution fashion. By evolving the programs of constructive methods, the finally obtained optimization program could beat certain human-crafted heuristics. OptiMUS (AhmadiTeshnizi et al., 2023), on the other hand, integrates hints about the target optimizer into the prompts to create and test new optimizers for numerical optimization problems. Differently, our LLaMoCo facilitate a novel paradigm: fine-tuning LLMs to generate advanced optimizer programs for numerical optimization in one round conversation at the instance level. To this end, we propose a stipulated optimization problem description format, a novel automated synthetic problem generation procedure and a large-scale benchmarking process to construct a comprehensive instruction set. To the best of our knowledge, all the aforementioned works focus on prompt engineering of pre-trained LLMs, and the area of fine-tuning general LLMs with optimization-domain knowledge remains unexplored.

# 3   LLaMoCo

LLaMoCo is the first instruction tuning framework for adapting general-purpose LLMs to generate instance-level optimizers. Operating on a code-to-code basis, it takes an optimization problem written with Python or LaTeX and generates a code implementation of a customized optimizer (as illustrated in Appendix F). Achieving such code-to-code flexibility encounters several unique challenges. First, we propose a novel synthesizing procedure which could automatically generate sufficient and diverse synthetic functions with/without constraints for instruction-tuning (Section 3.1). Next, to construct high-quality prompt-answer data for the instruction tuning, we propose an innovative and automated large-scale benchmarking process to facilitate effective answer labeling (Section 3.1). At last, based on the semantic alignment issue and data imbalance issue in the constructed instruction set, we design a novel two-phase tuning strategy to fine-tune general LLMs, as detailed in Section 3.2

## 3.1   CONSTRUCTION OF INSTRUCTION SET

**Task synthesis.** An optimization problem can be mathematically formulated as follows:

$$\min_x f(x), \quad g_i(x) \leq 0, i = 1, ..., M_g, \quad h_j(x) = 0, j = 1, ..., M_h \tag{1}$$

where $f(x)$ is the objective function, $g_i(\cdot)$ and $h_j(\cdot)$ denote $M_g$ inequality constraints and $M_h$ equality constraints respectively. Without loss of generality, we assume a minimization problem where the optimal solution $x^*$ attains the minimum objective value, adhering to all specified constraints.

**Optimization task synthesizing.** The first concern is generating a sufficient number of high-quality and diverse problem instances for instruction tuning (Sanh et al., 2022; Zhou et al., 2023). As it is impractical to gather all types of real-world optimization problems, we generate synthetic instances that represent various problem landscapes. Specifically, we collect a basic function set $F$ with various optimization problems and a basic constraint set $\Omega$ with various constraints from the well-known benchmarks (Boyd and Vandenberghe, 2004; Wu et al., 2017; Guo et al., 2023c). Following Mohamed et al. (2021), we synthesize a new objective function from $K$ basic functions in $F$ through two different paradigms as given by Equation (2): 1) *Composition*: a linear combination of the $K$ basic functions over the entire decision space, with each $w_i$ uniformly sampled from $[0, 1]$. 2) *Hybrid*: The decision vector $x$ is randomly decomposed into $K$ segments ($s_1$ to $s_K$). Each basic function operates on one segment, and the final objective function is their summation.

$$Composition: f(x) = \sum_{i=1}^{K} w_i \cdot f_i(x), \quad Hybrid: f(x) = \sum_{i=1}^{K} f_i(x[s_i]) \tag{2}$$

We then process each problem instance in three steps: 1) indicate the problem dimension $D$, the search bounds for each dimension (e.g., $-10 \leq x_i \leq 10$), and the number of basic functions $K$; 2) if $K = 1$, randomly select a basic function in $F$ as $f(x)$, otherwise, we apply *Composition/Hybrid* paradigm to synthesize $f(x)$; and 3) randomly select a group of constraints $\{\{g_i\}, \{h_j\}\}$ in $\Omega$. Note that step 3) is optional, as some optimization problems may not have constraints. In this work, we generate 3k problem instances without constraints, denoted as $P_{nc}$, and another 3k problem

instances with constraints, denoted as $P_c$. The complete set $P$ is the union of $P_{nc}$ and $P_c$, consisting of 6000 instances. These instances showcase different characteristics of global landscapes, including unimodal or multimodal, separable or nonseparable, and symmetrical or asymmetrical. They also exhibit various local landscape properties, such as distinct properties around different local optima, continuous everywhere yet differentiable nowhere, and optima situated in flattened areas. This guarantees that the generated instances comprehensively mirror various realistic problems.

**Knowledge gathering.** In our study, the term 'knowledge' refers to expertise in handling optimization problems, including selecting a well-performing optimizer and configuring its hyperparameters. To this end, we conduct exhaustive benchmarking to determine one effective optimizer for each instance $p \in P$. Concretely, we filter a wide range of optimizers from the literature (Stork et al., 2022; Zhan et al., 2022), competitions (Wu et al., 2017; Mohamed et al., 2021; Turner et al., 2021), and benchmarks (R.Turner and D.Eriksson, 2019; Guo et al., 2023c), selecting 23 optimizers that span various algorithm families, including Evolutionary Algorithms (e.g., GA (Holland, 1992; Clune et al., 2008; Wang et al., 2023), DE (Storn and Price, 1997; Xu et al., 2020; Biswas et al., 2021; Ye et al., 2023), PSO (Kennedy and Eberhart, 1995; Gong et al., 2015; Wu and Wang, 2022; Lu et al., 2023) and ES (Hansen and Ostermeier, 2001; Ros and Hansen, 2008; Hansen, 2009; He et al., 2020)), Bayesian Optimization (Snoek et al., 2012; Wang et al., 2020), Local Search strategies (Kirkpatrick et al., 1983; Van Laarhoven et al., 1987; Xiang et al., 1997; Fontes et al., 2023), and Numerical Optimization methods (Kraft, 1988; Conn et al., 2000; Powell, 2007; Bollapragada et al., 2018). To determine the most effective optimizer for each instance $p$, we employ a two-step process. Firstly, we perform a grid search to identify the best configuration for each optimizer on $p$ (conducted multiple runs to reduce the impact of variance). Subsequently, we select the optimizer that yields the best performance among all the configured optimizers. The selected optimizer and its configuration are implemented as a piece of Python code, denoted as $a_p$, serving as the knowledge of the desired optimizer's implementation for instance $p$. Refer to Appendix A and Appendix E.4 for details and correctness of the benchmarking process respectively.

**Instruction set construction.** So far we have obtained a problem set $P$ of 6000 optimization instances and their best-performing optimizer code. However, how to describe each instance $p \in P$ in a programming language has not been resolved. Common sense is that we human beings describe an optimization problem's mathematical formulation in computers by programming languages such as Python or LaTeX. However, even with such universal languages, different users created diverse code pieces to formulate the given problem due to their programming habits. This motivates us to augment each instance in $P$ to diverse problem descriptions written in either Python or LaTeX. Training on such augmented data could help LLaMoCo output consistent optimization code even if its inputs are different description versions of the same problem instance. To achieve this, we have conducted a survey among computer science students, and let them independently write problem descriptions for instances in $P$ by both Python and LaTeX. We analysed the collected scripts and found several major programming patterns. By using these patterns, we create $4 \sim 6$ Python or LaTeX problem descriptions for each instance. Due to the space limitation, we provide details about the found patterns in Appendix B. After the augmentation, for each problem instance $p \in P$, we first insert each description of it into the prompt template to attain $4 \sim 6$ text prompts $\{q_{1,p}, q_{2,p}, ...\}$. We then insert the selected best-performing optimization code for $p$ into the answer template to attain a text answer $a_p$. At last, for each $p$, we construct $4 \sim 6$ prompt-answer pairs $\{(q_{1,p}, a_p), (q_{2,p}, a_p), ...\}$. By repeatedly constructing prompt-answer pairs for all instances in $P$, we finally construct an instruction tuning set $\mathbb{I}$ comprising 32570 prompt-answer pairs.

### 3.2 TWO-PHASE INSTRUCTION TUNING

**Contrastive warm-up.** Given the constructed instruction set $\mathbb{I}$, although we could naively apply regular LM loss to train an LLM to fit prompt-answer pairs in $\mathbb{I}$, the training convergence and stability suffer from two cases: a) there are 6000 problem instances involved, and they can be categorized into 23 classes, where the class label is the optimizer achieving best performance on each instance. This means that there is a chance two different problems hold very similar answers. b) for two different problems, the difference in their programming language description might be very small. For example, consider two problems with the same objective function one with a constraint and the other without, the difference in their code descriptions is only one line. However, the answer for the one without constraint should be the DE algorithm and for the other should be SLSQP. The above two cases would confuse the LLM during the fine-tuning.

In fact, such an issue is endemic in code understanding scenarios (Guo et al., 2022) and a common practice is leveraging contrastive learning to align the latent representation for case a) and drag the latent representation apart for case b). In LLaMoCo, we follow the idea and design a contrastive warm-up process before instruction tuning. Concretely, to construct a mini-batch of training data, we first randomly select a prompt-answer pair $(q, a)$ from $\mathbb{I}$, where $q$ is regarded as the anchor prompt. For the positive sample, we randomly select a data pair $(q^+, a)$ with the same optimizer type (we have 23 types). For the negative

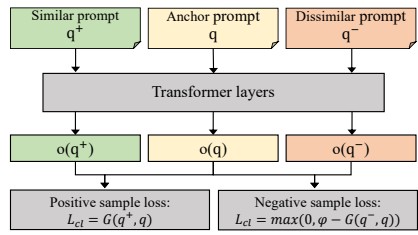

Figure 2: The contrastive warm-up .

sample, we randomly select a data pair $(q^-, a')$ with a different optimizer type. Then these three samples are applied to calculate the positive and negative sample loss, respectively. Specifically, for the decoder-only LLMs adopted for code generation tasks in this paper, we activate the Transformer layers (Vaswani et al., 2017) and regard the output embedding of the final self-attention block as a latent representation for a prompt. In LLaMoCo, we measure the distance between two prompts $q$ and $q'$, denoted as $G(q, q')$, by considering the cosine similarity between their latent representations $\vec{o}(q)$ and $\vec{o}(q')$:

$$G(q, q') = \frac{1}{2}\left(1 - \frac{\vec{o}(q) \cdot \vec{o}(q')}{\|\vec{o}(q)\| \, \|\vec{o}(q')\|}\right) \tag{3}$$

The distance $G(q, q') \in [0, 1]$. The contrastive loss of $q$ and $q'$, denoted as $L_{\mathrm{cl}}(q, q')$, is calculated as:

$$L_{\mathrm{cl}} = \begin{cases} G(q, q') & q' = q^+ \\ \max(0, \varphi - G(q, q')) & q' = q^- \end{cases} \tag{4}$$

where $\varphi$ is a margin parameter. By minimizing $L_{\mathrm{cl}}$, we could efficiently pull together the representations of two prompts which share the same desired optimizer yet have different forms, and vice versa. This contrastive phase is economic since we only consume a small number of epochs to warm up the fine-tuning of LLMs by $L_{\mathrm{cl}}$ and then instruction-tune the LLMs with the regular LM loss for next-token prediction (Wolf et al., 2019). We validate the effectiveness of this contrastive learning phase in Section 4.3.

**Balanced data sampling.** The instruction set $\mathbb{I}$ exhibits certain imbalance in the distribution of data. Notably, we observe that several optimizers dominate on thousands of problem instances, while the others only outperform on a few problem instances. Dealing with imbalanced data poses a challenge during the process of fine-tuning models (Batista et al., 2004; Zhao et al., 2023). To address the issue, we follow the example-proportional mixing strategy (Raffel et al., 2020) to re-balance the data distribution in $\mathbb{I}$. Each data pair $(q, a)$ is sampled with a probability $\rho$ as:

$$\rho(q, a) = \frac{1}{N_a \times N_{q,a}} \tag{5}$$

where $N_a$ denotes the number of optimizers in the gathered algorithm pool, $N_{q,a}$ denotes the number of instances whose desired optimizer is $a$. In this way, the number of sampled pairs dominated by each optimizer is approximately equal in each training epoch. Note that we apply this strategy in both the contrastive warm-up phase and the instruction tuning phase. The approach aids in avoiding biased training of the LLMs and enables them to effectively learn the knowledge from minority instances.

## 4 RESULTS AND DISCUSSIONS

### 4.1 EXPERIMENTAL SETUP

**Fundamental models & Training settings.** We adopt CodeGen-Mono (350M), Phi-2 (2.7B) and Code Llama (7B) as backbone models and fine-tune them on our instruction set. The reasons are two-fold: 1) these models show robust programming language reasoning and code generation ability, serving as a good starting point for the code-to-code scenario in our work; 2) the relatively small model size helps to reduce computational resources required for training and deploying. For generating the task set $P$, the range of the problem dimension is $[2, 50]$, and the number of components $K$ is randomly chosen from $[1, 5]$. We randomly split the instruction set $\mathbb{I}$ into a training set $\mathbb{I}_{\mathrm{train}}$ with 30k input-output pairs and a test set $\mathbb{I}_{\mathrm{eval}}$ with the rest examples. We leave other details in Appendix C.

Table 1: Results of different approaches in terms of **Code Error Rate (Err.)**, **Code Recovery Cost (Rec.)**, **Optimization Performance (Perf.)**, and **Computational Overhead (Comp.)** on the unconstrained problems ($\mathbb{I}_{eval}/P_c$), constrained problems ($\mathbb{I}_{eval}/P_{nc}$), both constrained and unconstrained problems ($\mathbb{I}_{eval}$), and realistic problems $\mathbb{I}_{real}$, where "-" denotes that the approach does not generate code (it follows a solution-to-solution paradigm).

| Testset | Metrics | Prompt for Solution | | Prompt for Optimizer | | | | | | Our LLaMoCo | | |
|---|---|---|---|---|---|---|---|---|---|---|---|---|
| | | OPRO | LMEA | CodeGen-Mono-350M | Phi-2-2.7B | DeepSeekMath-Instruct-7B | GPT-4 Turbo | Code Llama-7B | Llama2-70B | LLaMoCo-S | LLaMoCo-M | LLaMoCo-L |
| $\mathbb{I}_{eval}/P_c$ | Err. ↓ | - | - | 99.864% | 97.413% | 71.564% | 43.333% | 98.184% | 99.673% | 5.437% | **4.414%** | 4.697% |
| | Rec. ↓ | - | - | 80.234% | 72.242% | 13.483% | 9.942% | 67.857% | 62.232% | **9.684%** | 10.101% | 9.947% |
| | Perf. ↑ | 29.499% | 20.350% | 12.341% | 17.313% | 58.568% | 71.783% | 14.089% | 18.922% | 85.360% | **86.412%** | 85.810% |
| | Comp. ↓ | 115k | 249k | 1.9k | 2.1k | 2.1k | 3.4k | 1.7k | **1.5k** | 2.3k | 2.3k | 2.3k |
| $\mathbb{I}_{eval}/P_{nc}$ | Err. ↓ | - | - | 99.413% | 92.234% | 67.488% | 39.944% | 90.474% | 99.521% | **5.697%** | 6.130% | 5.977% |
| | Rec. ↓ | - | - | 78.341% | 54.156% | 12.145% | 16.463% | 44.938% | 49.202% | 11.861% | **10.443%** | 10.584% |
| | Perf. ↑ | 4.514% | 7.541% | 20.314% | 41.342% | 53.477% | 75.678% | 46.968% | 22.460% | 77.576% | 79.718% | **83.404%** |
| | Comp. ↓ | 115k | 249k | 2.1k | 2.2k | 2.1k | 3.5k | **2.0k** | **2.0k** | 2.5k | 2.5k | 2.5k |
| $\mathbb{I}_{eval}$ | Err. ↓ | - | - | 99.421% | 94.314% | 69.371% | 41.667% | 95.156% | 99.617% | 5.580% | **5.434%** | 5.509% |
| | Rec. ↓ | - | - | 79.371% | 63.314% | 12.518% | 13.072% | 57.001% | 55.717% | 10.826% | **10.349%** | 10.461% |
| | Perf. ↑ | 17.821% | 14.762% | 15.341% | 19.345% | 55.847% | 74.248% | 29.717% | 20.579% | 81.843% | 83.369% | **83.451%** |
| | Comp. ↓ | 115k | 249k | 2.0k | 2.1k | 2.1k | 3.5k | 1.9k | **1.7k** | 2.4k | 2.4k | 2.4k |

**Baselines.** We include two solution-to-solution approaches, OPRO (Yang et al., 2023) and LMEA (Liu et al., 2023a), which prompt pre-trained LLMs (e.g., GPT-4 Turbo) repeatedly to generate and improve solutions for the given problems. Compared to OPRO, LMEA additionally engineered its prompt with an explicit indication of using some evolutionary operators to let LLMs act as an evolutionary optimizer for performance boost. We also include six general LLMs for code generation, namely CodeGen-Mono-350M (Nijkamp et al., 2023), Phi-2-2.7B (Javaheripi et al., 2023), Code Llama-7B (Roziere et al., 2023), Llama 2-70B (Touvron et al., 2023), DeepSeedMath-Instruct-7B (Shao et al., 2024) and GPT-4 Turbo (Achiam et al., 2023). We prompt these three general LLMs with the same format as in our instruction set $\mathbb{I}$ to generate an optimizer for each problem instance. Note that we do not include the LLMs for algorithm design works such as EoH (Liu et al., 2024) into the comparison, since works in this line primarily address the combinatorial optimization problems but our LLaMoCo revolves around numerical optimization problems. The configurations of the baselines are set by default according to the corresponding references, and listed in Appendix C. Note that we do not include Algorithm Selection methods (Kerschke et al., 2019; Guo et al., 2024) for comparison since LLaMoCo generates complete optimizer source codes that not only specify the selected algorithm but also include the necessary implementation details, which is beyond the scope of standard algorithm selection. Besides, LLaMoCo performs hyper-parameter tuning as part of the optimization code generation process, providing a level of configurability that algorithm selection methods cannot achieve.

**Performance metrics.** When evaluating the performance of LLMs for optimization, we consider four metrics: 1) the *code error rate*, which indicates the proportion of problems for which the LLMs generate optimization codes with bugs (lower values are preferable); 2) the *code recovery cost*, which measures the proportion of lines of code that need to be corrected in order to fix the bugs in the erroneous codes (lower values are preferable); 3) the average *optimization performance* on the test problems (higher values are preferable), which is a min-max normalized term indicating the optimization results over the tested problem set; and 4) the average *computational overhead* for solving a problem, which is determined by the number of tokens used for both the input and output of LLMs (lower values are preferable). These four metrics could provide a comprehensive evaluation of existing baselines and our LLaMoCo in aspects of code generation robustness, optimization performance and runtime complexity. The detailed calculations can be found in Appendix D.

## 4.2 PERFORMANCE ANALYSIS

We use LLaMoCo-S(mall), -M(edium) and -L(arge) to denote the fine-tuned CodeGen-Mono (350M), Phi-2 (2.7B) and Code Llama (7B) models on $\mathbb{I}_{train}$, respectively.

**Performance on test sets.** First, we evaluate the performance of our fine-tuned LLMs and the competitors on three test sets, $\mathbb{I}_{eval}/P_c$, $\mathbb{I}_{eval}/P_{nc}$, and $\mathbb{I}_{eval}$ that represent the unconstrained task set, constrained task set, and the complete set mixing unconstrained and constrained tasks, respectively, each with 5 independent runs. The results are reported in Table 1, which show that: 1) The LLMs fine-tuned by our LLaMoCo framework consistently achieve superior performance, which validates that instruction tuning the general LLMs with moderate expert-level knowledge would gain substantial performance reinforcement in optimization. For example, LLaMoCo-L fine-tuned on the Code Llama (7B) demonstrate a performance boost from 29.717% to 81.843% on $\mathbb{I}_{eval}$.

2) Although LLaMoCo-S is fine-tuned from a relatively small fundamental model, it achieves competitive performance to those of LLaMoCo-M and LLaMoCo-L. This reveals a potential marginal effect in instruction tuning, since the data scale should match the model capacity. See Appendix E.3 for a detailed experiment where we provide an initial exploration on LLaMoCo's scaling law.

3) The solution-to-solution approaches OPRO and LMEA achieve unsatisfactory performance on our complex optimization task sets. Considering the tremendous tokens these approaches consume to solve one optimization problem through iteratively prompting solutions, both the efficacy and efficiency (as shown in the 'Perf.' and 'Comp.' rows of Table 1) of them require further improvement.

4) Among the six 'prompt for optimizer' models we compared, the GPT-4 Turbo dominates the others, which shows the power of a general-purpose LLM with high capacity. Nevertheless, it still underperforms our domain-specific LLaMoCo. Our models effectively reduce the error rates and the required recovery efforts for generating the codes of an optimizer through the instruction tuning. Meanwhile, note that the Code Llama (7B) model achieves better overall performance than the Llama 2 (70B) model in our experiments. The above observations validate that, although LLMs with larger capacity may show strong performance for solving general tasks, a smaller model could be sufficient to be fine-tuned as a domain-specific task solver.

5) Without specifically trained on our proposed $\mathbb{I}_{train}$, all general LLM baselines show high error rates when generating the optimizer program. This further outlines the advantages of instruction-tuning the general LLMs with LLaMoCo (with at most $6.13\%$ error rate for our three pre-trained models). We have to note that recent few-shot prompting researches (Madaan et al., 2022; Bareiß et al., 2022) indicate that the error rate could be decreased by providing the general LLMs with moderate examples as hints. We also conduct a comparison study between our LLaMoCo models and the three general LLMs enhanced by the few-shot prompting strategy (see Appendix E.1). Although the error rates of the three general LLM baselines are significantly reduced, the error rates are still over $10\%$. Besides, few-shot prompting for the general LLMs requires additional computational overheads and certain expertise for the examples.

6) We additionally use our LLaMoCo to fine-tune all models of codeGen series, from 350M to 7B. We present the optimization performance of these models before and after LLaMoCo's instruction tuning in Table 2. The results clearly validate the robust performance boosting ability of LLaMoCo for different model-size LLMs.

Table 2: Performance comparison across 350M $\sim$ 7B CodeGen-Mono models before and after LLaMoCo's instruction-tuning.

| Model | Model Size | | | |
|---|---|---|---|---|
| | 350M | 1B | 3B | 7B |
| CodeGen-Mono | 15.341% | 18.943% | 19.348% | 20.982% |
| LLaMoCo-CodeGen | **81.843%** | **82.541%** | **83.315%** | **83.513%** |

**Zero-shot performance on realistic problems.** To validate the generalization of LLaMoCo on intricate real-world scenarios, we compare models fine-tuned by our LLaMoCo and the other baselines on a wide range of realistic instances. Concretely, we select 8 realistic optimization tasks: **1) Haverly's Pooling** problem (Floudas and Pardalos, 1990) for allocating gas flow in pipeline transportation networks with minimum cost, which represents a linear-objective non-linear COP. **2) Multi-product batch plant** problem (Grossmann and Sargent, 1979) for scheduling operations of multiple products in a plant. **3) Robot gripper** problem (Osyczka et al., 1999) for controlling the robotic gripper to grab target objects. **4) Wind farm layout** problem (Wang et al., 2017) for optimizing the locations of wind turbines to maximize the total power output. **5) SOPWM** problem (Rathore et al., 2010) for regulating Medium-Voltage drives. **6) Protein docking** problem (Hwang et al., 2010) for optimizing the docking pattern to achieve a stable protein-protein complex. **7) HPO** problem (Arango et al., 2021) for finding optimal hyper-parameter settings for machine learning algorithm. **8) Neuroevolution** problem (Such et al., 2017) for optimizing neural networks to address various downstream tasks such as classification and control. These problems show diverse optimization challenges such as optimization with complex constraints, rugged objective landscapes, expensive evaluation, ill-conditioned objective landscapes, multimodality etc. We present the results in Table 3, where we provide the optimization performances of all baselines and our LLaMoCo. The results demonstrate the robust performance of LLaMoCo for optimization problems in our daily life. This generalization roots from the stipulated problem description we proposed in this paper, where either the synthetic problems or the realistic problems are represented by programming language hence share the semantic consistency, which helps the generalizability of LLaMoCo. We further validate the zero-shot performance of

LLaMoCo on a realistic problem collection proposed by Kumar et al. (2020), which covers 57 diverse real-world optimization scenarios. The results is provided in Appendix E.2.

Table 3: Performance comparison between LLaMoCo and other baselines on realistic problems. These problems cover various domains such as engineering, continuous control, AutoML, and scientific discovery, showing diverse optimization challenges.

| Methods | Realistic Problems | | | | | | | |
|---|---|---|---|---|---|---|---|---|
| | Haverly's Pooling | Multi-product batch plant | Robot gripper | Wind Farm Layout | SOPWM for 3-level inverters | Protein Docking | HPO | Neuroevolution |
| OPRO | 58.567% | 37.153% | 26.488% | 31.499% | 34.125% | 28.315% | 58.567% | 18.243% |
| LMEA | 46.852% | 40.256% | 34.786% | 29.457% | 36.445% | 17.342% | 46.852% | 19.423% |
| CodeGen-Mono-350M | 43.729% | 33.154% | 43.155% | 19.456% | 40.564% | 18.348% | 43.729% | 24.354% |
| Phi-2-2.7B | 50.345% | 40.782% | 49.486% | 34.498% | 46.557% | 40.348% | 50.345% | 26.487% |
| DeepSeekMath-Instruct-7B | 62.782% | 57.121% | 59.478% | 53.145% | 66.784% | 61.783% | 62.782% | 34.364% |
| GPT-4 Turbo | 63.487% | 67.158% | 58.586% | 58.489% | 76.447% | 74.284% | 63.487% | 37.145% |
| Code Llama-7B | 51.425% | 53.187% | 55.406% | 46.487% | 70.887% | 39.341% | 51.425% | 32.451% |
| Llama2-70B | 49.488% | 51.156% | 52.045% | 44.267% | 68.456% | 37.481% | 49.488% | 35.478% |
| LLaMoCo-S | 83.152% | 75.145% | 61.798% | 75.412% | 81.364% | 80.734% | **83.657%** | 57.364% |
| LLaMoCo-M | 80.468% | **78.851%** | **64.587%** | 75.654% | 82.457% | 81.044% | 80.468% | **58.145%** |
| LLaMoCo-L | **86.758%** | 76.148% | 62.891% | **77.364%** | **82.669%** | **81.532%** | 83.152% | 57.341% |

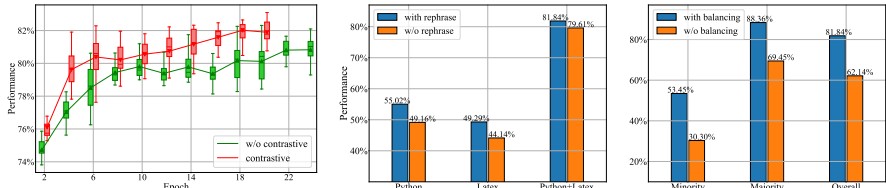

Figure 3: **Left**: Effectiveness of the contrastive warm-up strategy presented as the performance gains along the instruction tuning process. The proposed contrastive warm-up boosts both the learning efficiency and the learning effectiveness of LLaMoCo. **Middle**: Effectiveness of the diversity enhancement strategy. **Right**: Effectiveness of the balanced data sampling strategy.

Table 4: Comparison between LLaMoCo and the OpenAI models, in terms of **Code Error Rate (Err.)**, **Code Recovery Cost (Rec.)**, **Optimization Performance (Perf.)**, and **Computational Overhead (Comp.)** on the test set $\mathbb{I}_{\text{eval}}$.

| | | GPT-4 Turbo | GPT-4o | o1-mini | o1-preview | GPT-4 vector search | LLaMoCo-S | LLaMoCo-M | LLaMoCo-L |
|---|---|---|---|---|---|---|---|---|---|
| $\mathbb{I}_{\text{eval}}$ | Err. ↓ | 41.667% | 33.771% | **3.355%** | 4.107% | 9.336% | 5.580% | 5.434% | 5.509% |
| | Rec. ↓ | 13.072% | 14.405% | **10.299%** | 10.641% | 12.853% | 10.826% | 10.349% | 10.461% |
| | Perf. ↑ | 74.248% | 75.193% | 80.269% | 79.945% | 79.944% | 81.843% | **86.412%** | 85.810% |
| | Comp. ↓ | 3.5k | 3.6k | 4.1k | 4.1k | 7.1k | **2.4k** | **2.4k** | **2.4k** |

## 4.3 ABLATION STUDY

**Contrastive warm-up.** The contrastive warm-up phase in our proposed two-phase instruction tuning strategy (see Section 3.2) aims to reduce the cross-modal ambiguity by aligning the latent representations of different prompts that share the same desired optimizer (vice versa). We illustrate the performance gain curves on $\mathbb{I}_{\text{eval}}$ with or without the contrastive warm-up during the instruction tuning in the left of Figure 3, where LLaMoCo-S is applied as a showcase and the error bars indicate the variance across 5 training runs. The results show that incorporating such a contrastive warm-up strategy aids in accelerating the convergence of the subsequent instruction tuning. Furthermore, it is advantageous for the LLMs to generate accurate codes and enhance the overall performance. We refer to Appendix E.5 for all results on different LLaMoCo models and test problem sets.

**Diversity enhancement.** To improve the generalization of the fine-tuned LLMs in LLaMoCo, we enrich the task descriptions for each problem instance by augmenting the description of its objective function and constraints with Python or LaTeX codes of different writing styles. We illustrate the effect of this procedure in the middle of Figure 3 by showing the optimization performance of six LLaMoCo-S models trained on pure Python, pure LaTeX and Python+LaTeX data, with or without the diversity enhancement by rephrasing. The results show that providing multi-lingual descriptions of optimization problems significantly boosts the generalization performance, while rephrasing each description with multiple writing styles further enhances the final training results.

**Balanced data sampling.** In LLaMoCo, we address the imbalanced data distribution (caused by dominate optimizers) through performing example-proportional sampling on $\mathbb{I}_{\text{train}}$. To investigate its

effectiveness, we train two LLaMoCo-S models on $\mathbb{I}_{\mathrm{train}}$, with or without the data balancing strategy, respectively. The optimization performance of the two models is presented in the right of Figure 3, by separately considering the majority instances (which request the dominating optimizers), the minority instances (which request the others), and the overall instances of $\mathbb{I}_{\mathrm{eval}}$. The results consistently show that keeping a balanced training data distribution significantly boosts performance.

### 4.4 OPEN-ENDED DISCUSSION: IS GPT-4 A TRUE OPTIMIZATION EXPERT?

Considering the competitive performance of GPT-4, as shown in Table 1, we delve into whether GPT-4 can be deemed as a genuine optimization expert. Upon viewing the optimization codes generated by GPT-4 for both test and realistic problem sets, a noteworthy pattern emerges. GPT-4 consistently leans towards generating a specific numerical optimizer, SLSQP (Kraft, 1988), for almost all tested problems. While SLSQP is a classical solver for convex quadratic programming and is included in our chosen advanced optimizers, our benchmarking results identify that on a proportion of tested problems, it underperforms the others such as the Vanilla DE (Storn and Price, 1997).

To investigate further, we have conducted testing on a series of latest GPT models: GPT-4o, GPT o1-mini and GPT o1-preview. In particular, we also add a GPT-4 vector search baseline (OpenAI, 2023) which uses a vector search method to provide in-context enhancement for the GPT-4 model. We present the performance comparision in Table 4. From the results, we can observe that: a) o1 v.s. GPT-4o: indeed, o1 models achieve significantly lower coding errors than 4o model, demonstrating their robust coding enhancement. b) o1-mini v.s. LLaMoCo: on the one hand, the error rate of o1-mini is lower than our LLaMoCo, which originates from the black-box training of o1-mini on extremely large coding tasks. On the other hand, LLaMoCo, trained on a very small model, could achieve more optimization performance gain, with fewer tokens consumed. Besides, we look into the source codes generated by o1-mini as we have done for GPT-4 Turbo model in Section 4.4. It turns out that o1-mini also leans toward generating a particular optimizer, DE algorithm, for almost all tested problems. This further underscores the core motivation of LLaMoCo, which is exploring how to inject domain-specific knowledge into LLMs to adapt them for specific tasks. c) By providing GPT-4 Turbo with an example prompt-answer pair which is similar to the tested prompt, the error rate of the generated optimizer code significantly declines. d) However, such a prompting strategy consumes doubled tokens than directly prompting GPT-4 Turbo, which is inefficient considering our LLaMoCo only requires 2.4k tokens to achieve superior optimization performance. This underscores the importance of our LLaMoCo for adapting LLMs to solve optimization problems.

## 5 CONCLUSION

We introduce LLaMoCo, the first instruction-tuning framework to adapt general LLMs to function as expert-level systems to solve optimization problems. To achieve this, we meticulously construct an instruction set with more than 30k demonstration examples and then employ a novel two-phase instruction tuning strategy to fine-tune a series of LLMs. The results show that our models consistently outperform existing approaches. Notably, we observe that a relatively small LLM is sufficient to be tuned as an expert-level optimization code generator superior to GPT-4. As a pioneering work, LLaMoCo holds certain limitations. On the one hand, The current LLaMoCo has been fine-tuned on a dataset comprising 30k single-objective optimization problems, serving as a proof of concept for the framework's potential on larger-scale training. Hence, these models may not yet be fully equipped to handle all out-of-distribution problem types. Exploring LLaMoCo's applicability to other additional problem domains represents an exciting avenue for future research. On the other hand, beyond the current reliance on a labeled dataset for fine-tuning, we can see significant potential in further enhancing LLaMoCo's efficiency and generalization capabilities by integrating Alignment Tuning (AT), which not only promises to refine the model's performance with expert optimization knowledge but also ensures that the solutions it generates are more interpretable and aligned with human knowledge. Other future research directions include but not limited to: a) the potential application of the proposed contrastive warmup strategy to other domains outlines an interesting and open-ended future work. b) the design of the instruction dataset such as employing CoT (Wei et al., 2022b) to enhance the reasoning ability of the fine-tuned model, is very promising. c) incorporate LLaMoCo with effective multi-turn code optimization framework might be a promising way to further improve both the code correctness and the optimization performance of LLaMoCo. In a word, we propose LLaMoCo with the hope that this preliminary exploratory research could appeal to more researchers to explore the potential of LLMs as Optimizers.

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

# A BENCHMARKING FOR KNOWLEDGE GATHERING

## A.1 OPTIMIZER POOL AND THE USED ASSETS

To match each problem instance in the generated problem set $P$ with an appropriate optimizer with corresponding code implementation, we construct an optimizer pool $\Lambda$ which integrates 23 well-performing optimizers from various algorithm families. These selected optimizers can be divided into two groups considering their compatibility for constraint handling. We briefly list the two groups as below:

**Unconstrained group $\Lambda_{uc}$:** Simulated Annealing (Kirkpatrick et al., 1983), Vanilla PSO (Kennedy and Eberhart, 1995), Vanilla DE (Storn and Price, 1997), Dual Annealing (Xiang et al., 1997), SAMR-GA (Clune et al., 2008), SEP-CMA-ES (Ros and Hansen, 2008), BIPOP-CMA-ES (Hansen, 2009), DEAP-DE (Fortin et al., 2012), Vanilla BO (Snoek et al., 2012), GLPSO (Gong et al., 2015), MMES (He et al., 2020), LA-MCTS (Wang et al., 2020), MadDE (Biswas et al., 2021), sDMS-PSO (Wu and Wang, 2022), AMCDE (Ye et al., 2023), NSA (Fontes et al., 2023).

**Constrained group $\Lambda_c$:** SLSQP (Kraft, 1988), Trust-Constr (Conn et al., 2000), COBYLA (Powell, 2007), L-BFGS-B (Morales and Nocedal, 2011), HECO-DE (Xu et al., 2020), DTPSO (Lu et al., 2023), GA-TDX (Wang et al., 2023).

We benefit from open-source libraries, including DEAP (Fortin et al., 2012), PyPop7 (Duan et al., 2022), evosax (Lange, 2023), SciPy (Virtanen et al., 2020) and Scikit-Optimizer (Louppe and Kumar, 2016) etc., for the easy implementation of the selected optimizers. We list the codebases we adopt for the implementation of these optimizers and their licenses in Table 5. We note that the development and deployment of our framework strictly follow those licenses.

Table 5: Used assets and their licenses

| Asset | Codebase | License |
|---|---|---|
| DEAP-DE (Fortin et al., 2012)
Vanilla PSO (Kennedy and Eberhart, 1995) | DEAP (Fortin et al., 2012) | LGPL-3.0 License |
| SAMR-GA (Clune et al., 2008)
BIPOP-CMA-ES (Hansen, 2009)
Simulated Annealing (Kirkpatrick et al., 1983) | evosax (Lange, 2023) | Apache-2.0 license |
| SEP-CMA-ES (Ros and Hansen, 2008)
MMES (He et al., 2020)
LA-MCTS (Wang et al., 2020)
NSA (Fontes et al., 2023) | PyPop7 (Duan et al., 2022) | GPL-3.0 license |
| Dual Annealing (Xiang et al., 1997)
SLSQP (Kraft, 1988)
COBYLA (Powell, 2007) | SciPy (Virtanen et al., 2020) | BSD-3-Clause license |
| Vanilla BO (Snoek et al., 2012) | Scikit-Optimizer (Louppe and Kumar, 2016) | BSD-3-Clause License |
| GA-TDX (Wang et al., 2023)
Vanilla DE (Storn and Price, 1997)
MadDE (Biswas et al., 2021)
AMCDE (Ye et al., 2023)
HECO-DE (Xu et al., 2020)
GLPSO (Gong et al., 2015)
sDMS-PSO (Wu and Wang, 2022)
DTPSO (Lu et al., 2023) | advanced-global-optimizers
https://pypi.org/project/advanced-global-optimizers/ | MIT License |

## A.2 BENCHMARKING PROCESS

The benchmarking process aims to find an appropriate configured optimizer for each problem instance $p \in P$. To this end, for each optimizer $\Lambda_k \in \Lambda$, we span a grid-search configuration space $C_k$ based on its tunable hyper-parameters, which is listed in Table 6. Take DEAP-DE (Fortin et al., 2012) as an example, it has three hyper-parameters and each of them has 5 optional values (pre-defined by us). We hence span the $C_k$ of DEAP-DE as a configuration space comprising $5 \times 5 \times 5 = 125$ configurations, each denoted as $C_k^j$. We now establish the target of our benchmarking process:

$$\underset{\Lambda_k \in \Lambda, C_k^j \in C_k}{\arg\max} \quad \mathbb{E}\left[eval(p, \Lambda_k, C_k^j)\right]$$

where $p$ denotes the tested problem instance, $eval$ denotes the final optimization performance by calling $\Lambda_k$ with configuration $C_k^j$ to solve $p$, and $\mathbb{E}$ denotes the expectation of the optimization

performance, which is unbiased-estimated by 5 independent runs in this work. For constrained problems, we benchmark $\Lambda_c$, while for unconstrained problems we benchmark $\Lambda_{nc}$. We note that the benchmarking process for each problem instance may encounter execution failures, e.g., some optimizers in $\Lambda_c$ can not handle equality constraints, some optimizers in $\Lambda_{nc}$ are incompatible with non-convex problems, BO optimizers are extremely time-consuming on high-dimensional problems. When failures occur, the corresponding $eval(p, \Lambda_k, C_k^j)$ is set to 0. After benchmarking $\Lambda$ on $P$, we provide a configured optimizer $a(\Lambda_k, C_k^j)$, and the corresponding code implementation as the desired optimizer for each $p$.

Table 6: Configurations and the hyperparameter tuning settings of the optimizers.

| Type | Algorithm | Parameters | Search range |
|---|---|---|---|
| GA | SAMR-GA Clune et al. (2008) | NP | [10, 20, 50, 100, 200] |
| | | elite_ratio | 0.0 |
| | | sigma_init | [0, 0.5, 1] |
| | | sigma_meta | [1, 2, 3, 4, 5] |
| | | sigma_best_limit | [0.0001, 0.001, 0.1] |
| | GA-TDX Wang et al. (2023) | beta | [0.1, 0.2, 0.3, 0.4, 0.5] |
| | | gamma | [1, 3, 5, 7, 9] |
| | | m | 1e10 |
| | | NP | [10, 20, 50, 100, 200] |
| DE | Vanilla DE Storn and Price (1997) | NP | [10, 20, 50, 100, 200] |
| | | F | [0, 0.5, 0.9] |
| | | Cr | [0, 0.5, 0.9] |
| | | mutation | {best1, best2, rand2, current2rand, current2best, rand2best2} |
| | | bound | {clip, periodic, reflect, rand} |
| | DEAP-DE Fortin et al. (2012) | NP | [10, 20, 50, 100, 200] |
| | | F | [0.1, 0.3, 0.5, 0.7, 0.9] |
| | | Cr | [0.1, 0.3, 0.5, 0.7, 0.9] |
| | HECO-DE Xu et al. (2020) | $F_0$ | 0.5 |
| | | $Cr_0$ | 0.5 |
| | | $A_{rate}$ | [2, 4, 6, 8] |
| | | $H_m$ | [1, 3, 5] |
| | | $NP_m$ | 12 |
| | | $NP_{min}$ | 40 |
| | | lamda | [10, 20, 30, 40] |
| | | $n_0$ | [1, 2, 3] |
| | | gamma | [0.05, 0.1, 0.2] |
| | MadDE Biswas et al. (2021) | p | [0.09, 0.18, 0.27, 0.36] |
| | | $P_{qBX}$ | [0.01, 0.1, 0.2, 0.3, 0.5] |
| | | $F_0$ | 0.2 |
| | | $Cr_0$ | 0.2 |
| | | $A_{rate}$ | [1.3, 1.8, 2.3, 2.8 ,3.3] |
| | | $H_m$ | [5, 10 ,15, 20] |
| | | $NP_m$ | [2, 4, 6, 8] |
| | | $NP_{min}$ | 4 |
| | AMCDE Ye et al. (2023) | $F_0$ | 0.2 |
| | | $A_{rate}$ | [1.6, 2.1, 2.6, 3.1, 3.6] |
| | | $H_m$ | [5, 10, 15, 20] |
| | | $NP_m$ | [3, 6, 9] |
| | | $NP_{min}$ | 4 |
| | | Gn | 5 |
| | | $pbc_1$ | [0.4, 0.5, 0.6] |
| | | $pbc_2$ | [0.4, 0.5, 0.6] |
| | | pw | [0.1, 0.2, 0.3] |
| | | pr | [0.005, 0.01, 0.05] |
| | | $pls_{succ}$ | 0.1 |
| | | $pls_{fail}$ | 0.0001 |
| PSO | Vanilla PSO Kennedy and Eberhart (1995) | NP | [10, 20, 50, 100, 200] |
| | | $phi_1$ | [1, 2, 3] |
| | | $phi_2$ | [1, 2, 3] |
| | GLPSO Gong et al. (2015) | pm | [0.01, 0.1, 0.2] |
| | | NP | [10, 20, 50, 100, 200] |
| | | nsel | 10 |
| | | w | 0.7298 |
| | | $c_1$ | 1.49618 |
| | | sg | 7 |
| | | rho | [0.1, 0.2, 0.3] |
| | sDMS-PSO Wu and Wang (2022) | w | [0.729, 0.271, 0.5] |
| | | NP | [33, 66, 99, 198] |
| | | $c_1$ | [1.49445, 3., 0.75] |
| | | $c_2$ | [1.49445, 3., 0.75] |
| | | m | [1, 3, 5] |
| | | R | [5, 10, 15] |
| | | LP | [5, 10, 15] |

**Table 6 continued from previous page**

| Type | Algorithm | Parameters | Search range |
|------|-----------|------------|--------------|
| | | LA | 8 |
| | | L | 100 |
| | | L_FEs | 200 |
| PSO | DTPSO
Lu et al. (2023) | p | [0.1, 0.5, 0.9] |
| | | sigma | [0.25, 0.5, 0.75] |
| | | gamma | [0.25, 0.5, 0.75] |
| | | $u_1$ | [0, 0.5] |
| | | $u_2$ | [0, 0.5] |
| | | $c_{1,1}$ | [0, 1.711897] |
| | | $c_{1,2}$ | [0, 1.711897] |
| | | $c_{2,1}$ | [0, 1.711897] |
| | | $c_{2,2}$ | [0, 1.711897] |
| | | ws | 0.9 |
| | | we | 0.4 |
| | | $NP_{init}$ | [50, 100, 200] |
| | | radius | [0.05, 0.1, 0.2] |
| ES | SEP-CMA-ES
Ros and Hansen (2008) | n_individuals | [10, 20, 50, 100] |
| | | c_c | [1, 2, 3, 4, 5] |
| | | sigma | [0.1, 0.3, 0.5] |
| | BIPOP-CMA-ES
Hansen (2009) | NP | [10, 20, 50, 100] |
| | | elite_ratio | [0.2, 0.5, 0.7] |
| | | sigma_init | 1 |
| | | mean_decay | 0 |
| | | min_num_gens | [10, 30, 50] |
| | | popsize_multiplier | [1, 2, 3, 4, 5] |
| | MMES
He et al. (2020) | a_z | [0.05, 0.1, 0.2] |
| | | c_s | [0.1, 0.3, 0.5] |
| | | ms | [2, 4, 6] |
| | | n_individuals | [25, 50, 100] |
| | | n_parents | [25, 50, 100] |
| | | sigma | [0.1, 0.3, 0.5] |
| BO | Vanilla BO
Snoek et al. (2012) | acq_func | [LCB, EI, PI, gp_hedge, EIps, PIps] |
| | | n_initial_points | [5, 10, 20] |
| | | initial_point_generator | [random, sobol, halton, hammersly, lhs] |
| | LA-MCTS
Wang et al. (2020) | n_individuals | [10, 20, 50, 100] |
| | | c_e | [0.01, 0.05, 0.1] |
| | | leaf_size | [10, 20, 30, 40, 50] |
| LS | Simulated Annealing
Kirkpatrick et al. (1983) | NP | [10, 20, 50, 100, 200] |
| | | sigma_init | [0.1, 0.3, 0.5] |
| | | sigma_decay | 1 |
| | | sigma_limit | [0.01, 0.05, 0.1] |
| | | temp_init | 1 |
| | | temp_limit | 0.1 |
| | | temp_decay | [0.9, 0.99, 0.999] |
| | | boltzmann_const | [1, 5, 10] |
| | Dual Annealing
Xiang et al. (1997) | initial_temp | [523, 5230, 50000] |
| | | visit | [1.62, 2.62, 3.62] |
| | | restart_temp_ratio | [2e-5, 2e-3, 2e-1] |
| | NSA
Fontes et al. (2023) | sigma | [0.1, 0.3, 0.5] |
| | | schedule | [linear, quadratic] |
| | | n_samples | [10, 20, 50, 100, 200] |
| | | rt | [0.9, 0.99, 0.999] |
| NO | SLSQP
Kraft (1988) | eps | [1e-12, 1e-10, 1e-8, 1e-6, 1e-4] |
| | Trust-Constr
Conn et al. (2000) | initial_tr_radius | [0.5, 1, 1.5, 2] |
| | | initial_constr_penalty | [0.5, 1, 1.5, 2] |
| | | factorization_method | [equality_constrained_sqp, tr_interior_point] |
| | COBYLA
Powell (2007) | rhobeg | [0.5, 1, 1.5, 2] |
| | L-BFGS-B
Morales and Nocedal (2011) | maxcor | [5, 10, 15, 20] |
| | | eps | [1e-12, 1e-10, 1e-8, 1e-6, 1e-4] |

### A.3 DATA QUALITY CONTROL

To ensure the dataset include only solvable and non-trivial problems, for those problems without constraints, the composition and hybrid construction on the base functions follows the procedure of IEEE CEC 2021 Single-objective Competition, where the optimum and optimal objective value of the constructed function are closed-form. By adding the rotation and shift to the optimum, we change the landscape to be optimized to make the solution non-trivial. For the constrained problems, we additionally run optimizers in our algorithm pool which specialize at solving constrained problems to optimize the constructed problem instances for multiple times (50 runs) to ensure the problem instance is without constraint conflict and hence solvable.

## B DETAILS OF DATA AUGMENTATION

It is a common practice to augment the training data for boosting the generalization performance in recent LLMs works (Sanh et al., 2022; Wei et al., 2022a; Chung et al., 2022). In LLaMoCo, we alter different writing styles of a problem's definition to generate moderate diverse prompts for each problem instance generated in $P$. For the different writing styles, we conducted a survey among university students majoring in computer science, inviting them to write diverse Python or LaTeX code that they believe is correct for defining the given problem instances in their own coding styles. From all of the collected scripts, we choose 50 most representative ones for analysis. After systematic statistics, we have empirically summarized several writing patterns, which we believe could approximately represent the major writing patterns of different users. Based on these different patterns, for each problem instance $p \in P$, we can obtain moderate rephrased versions for its objective function and constraints written by either Python or LaTeX code. We showcase the found patterns on a toy Katsuura problem which holds the formulation as:

$$Minimize: \quad f(x) = \frac{10}{D^2} \prod_{i=1}^{D} \left( 1 + i \sum_{j=1}^{32} \frac{\left| 2^j x_i - \text{round} \left( 2^j x_i \right) \right|}{2^j} \right)^{\frac{10}{D^{1.2}}} - \frac{10}{D^2}, X \in R^D$$

For LaTeX patterns, we found three different writing styles from the 50 scripts, which differ from each other mainly based on the laws of arithmetic, e.g., commutative law, distributive law and associative law. We illustrate some different LaTeX codes for our toy problem in Figure 4.

```
$\begin{aligned}
Minimize:\quad &f(x) = \frac{10}{D^2}
\prod_{i=1}^D\left(1+i \sum_{j=1}^{32}
\frac{\left|2^j x_i-\operatorname{round}\left(2^j
x_i\right)\right|}{2^j}\right)^{\frac{10}{D^{1
2}}}-\frac{10}{D^2} , X\in R^{D}\\
\end{aligned}$
```
```
$\begin{aligned}
Minimize:\quad &f(x) = \frac{10}{D^2}
\left[\prod_{i=1}^D\left(1+i \sum_{j=1}^{32}
\frac{\left|2^j x_i-\operatorname{round}\left(2^j
x_i\right)\right|}{2^j}\right)^{\frac{10}
{D^{12}}}-1\right] , X\in R^{D}\\
\end{aligned}
```
```
$\begin{aligned}
Minimize:\quad &f(x) =\frac{10\prod_{i=1}^D
\left(1+i \sum_{j=1}^{32} \frac{\left|2^j x_i-
\operatorname{round}\left(2^j x_i\right)\right|}
{2^j}\right)^{\frac{10}{D^{12}}}-10}{D^2} , x
= \left({x_1,x_2,...,x_D}\right)\\
\end{aligned}$
```

Figure 4: Three writing styles in LaTeX of the toy problem.

For Python patterns, the testees show different coding preferences on the writing styles of the objective functions and the constraints, e.g., some may prefer using temporary variables to store interim calculation results, some leverage *numpy* to facilitate matrix operations while others use a *for loop*, some may encapsulate the calculation details into a functional module etc. In Figure 5 we list some of these writing styles on the toy problem.

```
D = np.shape(x)[-1]
temp1 = np.power(D, 1.2)
temp2 = np.repeat(np.power(np.ones((1, 32)) * 2,
np.arange(1, 33)), x.shape[0], 0)
temp3 = np.ones(x.shape[0])
for i in range(D):
    temp4 = temp2 * np.repeat(x[:, i, None], 32, 1)
    temp5 = np.sum(np.fabs(temp4 -
np.floor(temp4 + 0.5)) / temp2, -1)
    temp3 *= np.power(1 + (i + 1) * temp5, 10 /
temp1)
temp6 = 10 / D / D
result = temp3 * temp6 - temp6
```
```
D = np.shape(x)[-1]
result = np.zeros(x.shape[0])
for i in range(x.shape[0]):
    result[i] = 10 / (D ** 2)
    for j in range(D):
        round_x = 0
        for k in range(32):
            round_x += np.abs(2**(k+1) * x[i][j] -
np.round(2**(k+1) * x[i][j])) / (2**(k+1))
        result[i] *= np.power(1 + (j + 1) * round_x,
10 / (np.power(D, 1.2)))
result[i] -= 10 / (D ** 2)
```
```
def f1(x):
    D = np.shape(x)[-1]
    temp1 = np.power(D, 1.2)
    temp2 = np.repeat(np.power(np.ones((1, 32)) *
2, np.arange(1, 33)), x.shape[0], 0)
    temp3 = np.ones(x.shape[0])
    for i in range(D):
        temp4 = temp2 * np.repeat(x[:, i, None], 32,
1)
        temp5 = np.sum(np.fabs(temp4 -
np.floor(temp4 + 0.5)) / temp2, -1)
        temp3 *= np.power(1 + (i + 1) * temp5, 10 /
temp1)
    return temp3
D = np.shape(x)[-1]
temp1 = f1(x)
temp2 = 10 / D / D
result = temp1 * temp2 - temp2
```

Figure 5: Three writing styles in Python of the toy problem.

## C   DETAILS IN EXPERIMENTS

**Training settings.** For generating the task set $P$, the problem dimension $D$ for each $p_i$ is randomly chosen from $[2, 50]$, and the number of components $K$ is randomly chosen from $[1, 5]$. We randomly split the instruction set $\mathbb{I}$ into a training set $\mathbb{I}_{\text{train}}$ with 30k input-output pairs and a test set $\mathbb{I}_{\text{eval}}$ with the rest examples. We leave other details in Appendix. For our two-phase instruction tuning, we deploy 5 epochs of contrastive warm-up and 20 epochs of instruction tuning for all fundamental models. Specifically, we first apply *SGD* (Amari, 1993) with a fixed learning rate $5 \times 10^{-4}$ in the contrastive warm-up phase, alongside $\varphi = 0.3$. Then, we apply *AdamW* (Loshchilov and Hutter, 2019) to optimize the LLMs in the instruction tuning phase. During the initial 1k iterations, the learning rate gradually increases from 0 to $5 \times 10^{-4}$ in a linear manner. Subsequently, it decreases to 0 according to a cosine schedule. The batch size in both phases is set to 4. Note that we fine-tune the CodeGen-Mono (350M) with full parameters, but apply LoRA (Hu et al., 2022) to fine-tune the larger Phi-2 (2.7B) and Code Llama (7B) models, with the rank $r = 8$, scaling factor $\alpha = 32$, and a dropout rate of 0.05. All experiments are performed on a platform with an Intel(R) Xeon(R) Gold 6348 CPU, 504GB RAM and a Nvidia A800 (80GB) GPU. Upon the settings, the training duration for CodeGen is one day, whereas Phi-2 and Code Llama require 2.5 days and 4 days, respectively.

**Settings of the baseline.** The general LLM baselines we considered in the comparison are configured by their default settings . In particular, we adopt a temperature of 0, a maximum token length of 2048, and the basemodel $codellama - 7b - instruct$ for Code Llama-7B, which follows the reported settings for their pass@1 evaluation on MBPP and HumanEval benchmark (see Table 2 in their paper Roziere et al. (2023)). Besides, the configurations for Llama2-70B baseline is with a temperature of 0.1, a maximum token length of 2048, and the basemodel $llama - 70b - chat$, which aligns with the settings for MBPP and HumanEval benchmark (Table 21 in their paper (Touvron et al., 2023)). To the best of our knowledge, the MBPP and the HumanEval benchmark are the most related code generation tasks to the optimization program generation task of LLaMoCo. We directly call the corresponding APIs (see `https://docs.llama-api.com/essentials/chat`) for the evaluation in our paper. As for GPT-4 Turbo, since there are limited literature discussing the code generation ability of GPT-4 Trubo, we adopt the default settings with a temperature of 1, a maximum token length of 2048, and the basemodel $gpt - 4 - 1106 - preview$ of the OpenAI API (see `https://platform.openai.com/docs/guides/text-generation/chat-completions-api`).

We would also clarify the configurations for OPRO and LMEA. Following the comparison settings stated at the LMEA paper (Liu et al., 2023a), the population size $N$ is set to 16, the maximum generation number $G$ is set to 250, for both OPRO and LMEA, with the backbone LLM as $gpt - 4 - 1106 - preview$ version of GPT-4 Turbo. The major difference of these two approaches lies on the concrete temperature settings. While OPRO reported 1.0 as the best performing temperature (see Section 5.1 of their paper (Yang et al., 2023)), LMEA (Liu et al., 2023a) adaptively adjust the temperature along the optimization horizon: with initial temperature as 1.0, which will increase by 0.1 if it fails to find a better solution for every 20 generations.

**Homogeneous batch sampling.** We further apply a homogeneous batch sampling strategy at the instruction tuning phase to reinforce the alignment of the different rephrasing version prompts for a problem $p \in P$. Concretely, we force the LLMs to sample data pairs which come from the same problem instances in a mini-batch. We observe consistent boosts in the training of LLaMoCo-S, LLaMoCo-M and LLaMoCo-L. By presenting the LLMs with a batch of homogeneous samples, they can learn patterns specific to these cross-modal prompts data more effectively.

**Batch size.** We would clarify that due to the resource limitation, all of our experiments are run on an NVIDIA A800 GPU. When we train the CodeGen-Mono (350M), the batch size is 4 for both phases in our two-phase learning strategy. However, for one A800, Phi-2 (2.7B) and Code Llama (7B) are too large to include a batch of 4 samples, even if we adapt LoRA for them. For Phi-2, the batch size is 3 and 2 for each learning phase, while 3 and 1 for Code Llama.

## D   CALCULATION OF EXPERIMENTAL STATISTICS

To provide a thorough evaluation on the LLMs fine-tuned by our LLaMoCo and the other approaches, for a group of $N_p$ problem instances, we first leverage the optimization programs generated by each LLM to optimize them, for 5 independent runs. Then we calculate the average error rate, recovery

cost, optimization performance and computational cost of an approach as the performance metrics of overall performance. The calculation details of these four performance metrics in our experimental results are as follows:

**Error rate (Err.)** The robustness of the generated optimization program is a very important index for quality of service (QoS). We measure the robustness by the proportion of error programs generated by an LLM, named as error rate. For each instance, we use the optimization program generated by an LLM (ours or the others) to optimize that instance for $5$ independent runs. We count the number of the generated programs which encounter compilation error or runtime error when being executed, denoted as $N_{err}$ (every single run on each instance is counted). Then the error rate of an approach on the tested instances is calculated as $\frac{N_{err}}{5 \times N_p}$.

**Recovery cost (Rec.)** While an optimization program may encounter compilation error or runtime error, we observe from our experiments that a certain proportion of the error programs could be repaired and recovered. We provide a metric named recovery cost to measure the efforts required to repair the generated programs. Concretely, during the test time, if the optimization program generated by an LLM was non-functional, we would give the tested LLM an additional turn of conversation to refine the errors of the generated optimization code. Concretely, we construct a prompt: for an optimization program $a_j$, we denote the number of lines in it as $L^{(j)}$, and the number of lines that need to be repaired as $L_{err}^{(j)}$. Then the recovery cost for $a_j$ is $r_j = \frac{L_{err}^{(j)}}{L^{(j)}}$, and the recovery cost considering all $N_{err}$ error programs is calculated as $\frac{\sum_{j=1}^{N_{err}} r_j}{N_{err}}$. For the case that the generated optimization code is still erroneous after the self-refine process, we set the performance of that LLM on that optimization problem as $0$.

**Optimization performance (Perf.)** We measure the optimization performance of an approach by a min-max normalized objective value descent. Concretely, we first estimate an optimal objective value $f_i^*$ for $i$-th problem instance, which can be easily obtained from our benchmarking process (achieved best objective value). For the given approach, we denote the performance on the $i$-th problem instance in $j$-th run as a min-max normalized term $w_{i,j} = 1 - \frac{f_{i,j}^* - f_i^*}{f_{i,j}^0 - f_i^*}$, where $f_{i,j}^0$ is the best objective value of the solutions initialized by the optimizer on solving the $i$-th problem instance in $j$-th run, and $f_{i,j}^*$ is the corresponding best objective the optimizer finds. We have to note that if the optimization code generated is still non-functional after the repairing process above, we assign a performance value of $0$ for that run. At last, the overall average optimization performance of the given approach on the $N_p$ instances can be calculated as follows: $\frac{\sum_{i=1}^{N_p} \sum_{j=1}^{5} w_{i,j}}{5 \times N_p}$.

**Computational overhead (Comp.)** Measuring the computational overhead by the wall-time complexity of an LLM-based approach is impractical since some of the LLMs only provide API for users. The network communication budget through calling the APIs would bias the ground results. We instead count the average number of tokens (input+output) consumed by an approach for solving a problem instance over the test runs.

# E ADDITIONAL EXPERIMENTAL RESULTS

## E.1 COMPARISON WITH FEW-SHOT PROMPTING STRATEGY

We conduct further investigation on the performance of the three general LLM baselines under the few-shot prompting setting, which would make the comparison more rigorous and convincing. Concretely, we test each baseline for 5 runs, on the test set $\mathbb{I}_{eval}$. For each tested instance, 1 or 2 few-shot prompts are randomly chosen as hints from $\mathbb{I}_{train}$ (it depends the pre-defined maximum context length of the corresponding LLM). We report the four average performance metrics of the original three general LLM baselines, their few-shot enhanced versions and our LLaMoCo-L in Table 7. The results show that few-shot prompting strategy does introduce performance improvement for the baselines. In particular, by showing the baselines with some prompts as hints, the Error rate and the Recovery cost are significantly reduced. However, due to the provided few-shot example prompts, it would require more computational overheads (Comp.).

Table 7: Comparison between our LLaMoCo models and the few-shot prompting enhancement of the general LLM baselines, in terms of **Code Error Rate (Err.)**, **Code Recovery Cost (Rec.)**, **Optimization Performance (Perf.)**, and **Computational Overhead (Comp.)** on the test set $\mathbb{I}_{\text{eval}}$.

| Testset | Metrics | Prompt for Optimizer | | | Prompt for Optimizer (few-shot) | | | Our LLaMoCo |
|---------|---------|-----------|--------------|-----------|------------------------|--------------------------|-------------------------|------------|
| | | GPT-4 Turbo | Code Llama-7B | Llama2-70B | GPT-4 Turbo (few-shot) | Code Llama-7B (few-shot) | Llama2-70B (few-shot) | LLaMoCo-L |
| $\mathbb{I}_{\text{eval}}$ | Err. ↓ | 41.667% | 95.156% | 99.617% | 10.546% | 15.235% | 10.235% | **5.509**% |
| | Rec. ↓ | 13.072% | 57.001% | 55.717% | 8.423% | 29.445% | **7.456**% | 10.461% |
| | Perf. ↑ | 74.248% | 29.717% | 20.579% | 76.568% | 45.775% | 37.456% | **83.451**% |
| | Comp. ↓ | 3.5k | 1.9k | **1.7k** | 7.0k | 6.1k | 6.5k | 2.4k |

Besides, we would also clarify that the three baseline LLMs (GPT-4 Turbo, Code Llama-7B and Llama-70B) have certain code generation ability and code semantics understanding ability (according to their papers or technical reports). However, these baselines fall short when compared to our LLaMoCo models, highlighting the core motivation behind LLaMoCo: to embed expert-level optimization knowledge within LLMs, thereby enhancing their performance in specific domains. While few-shot prompting can potentially improve LLMs' understanding of tasks, it necessitates users to supply expert-level knowledge explicitly in their prompts (for example, instructing the use of the Gurobi package for solving a mixed-integer programming problem). This requirement contradicts the fundamental aim of our work, which is to provide users with a solution that demands minimal knowledge of optimization to effectively address their tasks.

### E.2   ADDITIONAL ZERO-SHOT EVALUATIONS

We additionally test our trained model on the realistic problem collection where we sample the former six problems in Table 3. This collection is proposed by Kumar et al. Kumar et al. (2020). It contains 57 real-world constrained optimization problems collected from a diverse range of engineering scenarios. We present the comparison results (average on all 57 problems) of our LLaMoCo and the other baselines in Table 8. The results further validate the effectiveness and superior performance of our LLaMoCo.

Table 8: Zero-shot performance of different approaches in terms of **Code Error Rate (Err.)**, **Code Recovery Cost (Rec.)**, **Optimization Performance (Perf.)**, and **Computational Overhead (Comp.)** on realistic problems. ($\mathbb{I}_{\text{real}}$), where "-" denotes that the approach does not generate code (it follows a solution-to-solution paradigm).

| Testset | Metrics | Prompt for Solution | | Prompt for Optimizer | | | | | | Our LLaMoCo | | |
|---------|---------|------|------|--------------------|----------|-------------------------|-------------|---------------|------------|-----------|-----------|-----------|
| | | OPRO | LMEA | CodeGen-Mono-350M | Phi-2-2.7B | DeepSeekMath-Instruct-7B | GPT-4 Turbo | Code Llama-7B | Llama2-70B | LLaMoCo-S | LLaMoCo-M | LLaMoCo-L |
| $\mathbb{I}_{\text{real}}$ | Err. ↓ | - | - | 99.487% | 98.131% | 68.921% | 40.148% | 99.344% | 99.473% | 5.984% | 5.479% | **5.359**% |
| | Rec. ↓ | - | - | 81.166% | 58.546% | 15.470% | 16.791% | 58.101% | 59.189% | 10.648% | 10.486% | **10.198**% |
| | Perf. ↑ | 31.832% | 27.549% | 26.477% | 33.460% | 58.044% | 61.468% | 49.481% | 47.146% | 84.462% | 87.279% | **88.135**% |
| | Comp. ↓ | 253k | 298k | 2.1k | 2.1k | 2.1k | 3.6k | 2.0k | **1.9k** | 2.6k | 2.6k | 2.6k |

### E.3   IMPACT OF DATASET SIZE

To in-depth analyse LLaMoCo's scaling law, we chose the base model of our LLaMoCo-S, which is the CodeGen family (350M 1B, 3B, 7B), for the experiment. Further, we construct four training set with different sizes (1k, 5k, 15k, 30k). We presents the optimization performance of these 16 trained models on the test set $\mathbb{I}_{eval}$ in Table 9.

Table 9: Optimization performance comparison between different model sizes and dataset sizes.

| model/data | 1k | 5k | 15k | 30k |
|------------|--------|--------|---------|----------|
| 350M | 47.260% | 66.661% | 80.306% | 81.843% |
| 1B | 46.799% | 67.829% | 81.783% | 82.541% |
| 3B | 47.131% | 68.492% | 82.501% | 83.315% |
| 7B | 45.645% | 70.147% | 82.966% | **83.513%** |

The results above provide several key observations: a) when data size is very small (1k), increasing model size would not obtain any performance gain, which possibly indicates overfitting. b) for all

model sizes, increasing the dataset size could introduce performance gain consistently. c) In summary, both the model size and the dataset size could influence the final performance of LLaMoCo.

### E.4 ABLATION ON GRID SEARCH GRANULARITY

We conduct two additional benchmarking processes, with half and double granularity of our original setting. For example, if a hyper-parameter holds four optional values in our setting: [0.2, 0.4, 0.6, 0.8], half granularity denotes [0.2, 0.8], double granularity denotes [0.1, 0.2, 0.3, 0.4, 0.5, 0.6, 0.7, 0.8], etc. We present the averaged optimization performance of the most effective optimizers on our problem set searched by these two granularities, normalized by our original granularity's performance, as well as the averaged searching wall time for one problem instance in Table 10.

Table 10: The averaged optimization performance normalized by our original granularity's performance and averaged searching wall time for one problem instance of different grid search granularities.

|             | half    | our setting | double    |
|-------------|---------|-------------|-----------|
| Performance | 71.793% | 1           | 102.344%  |
| wall time   | 6s      | 211s        | 6379s     |

The results reveal an evident tradeoff between the searching effectiveness and the searching efficiency of different grid search granularities. The searching wall time increases exponentially since there are 4-5 hyper-parameters in an optimizer. However, the performance improvement obtained by spending so much computational resources is only 2.344%. This result validates the selection appropriateness of our grid search granularity.

### E.5 ADDITIONAL CONTRASTIVE WARM-UP ABLATION RESULTS

In this section we present the performance gain curves of LLaMoCo-S, LLaMoCo-M, LLaMoCo-L on three test sets $\mathbb{I}_{eval}/P_c$, $\mathbb{I}_{eval}/P_{nc}$ and $\mathbb{I}_{eval}$ in Figure 6, in which we can observe consistent learning enhancement through introducing our proposed contrastive learning warmup.

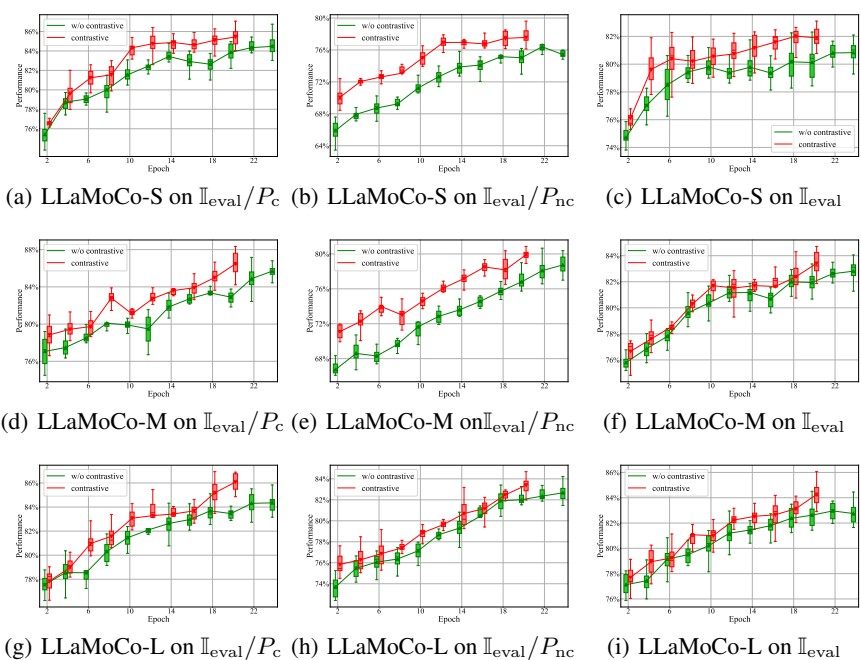

(a) LLaMoCo-S on $\mathbb{I}_{eval}/P_c$    (b) LLaMoCo-S on $\mathbb{I}_{eval}/P_{nc}$    (c) LLaMoCo-S on $\mathbb{I}_{eval}$

(d) LLaMoCo-M on $\mathbb{I}_{eval}/P_c$    (e) LLaMoCo-M on $\mathbb{I}_{eval}/P_{nc}$    (f) LLaMoCo-M on $\mathbb{I}_{eval}$

(g) LLaMoCo-L on $\mathbb{I}_{eval}/P_c$    (h) LLaMoCo-L on $\mathbb{I}_{eval}/P_{nc}$    (i) LLaMoCo-L on $\mathbb{I}_{eval}$

Figure 6: The performance gain curves of LLaMoCo-S, LLaMoCo-M, LLaMoCo-L on three test sets $\mathbb{I}_{eval}/P_c$, $\mathbb{I}_{eval}/P_{nc}$ and $\mathbb{I}_{eval}$ with and without contrastive learning.

## F    EXAMPLE INPUT-OUTPUT OF THE FINE-TUNED MODEL

### F.1    SYNTHETIC UNCONSTRAINED EXAMPLE

We showcase the prompt and the generated optimization program (Figure 7) of a synthetic problem instance without constraints, which has the following formulation:

$$Minimize: \quad f(x) = \sum_{i=0}^{1} W_i f_i(z), z = \mathbf{M}^{\mathrm{T}}x, x \in R^D, \mathbf{M} \in R^{D \times D}$$

$$Where: \quad f_0(\mathbf{x}) = -20 \exp\left(-0.2\sqrt{(1/D)\sum_{i=1}^{D} x_i^2}\right) - \exp\left((1/D)\sum_{i=1}^{D} \cos(2\pi x_i)\right) + 20 + e$$

$$f_1(\mathbf{x}) = \sum_{i=1}^{D}\left(\sqrt{|x_i|} + 2sin(x_i^3)\right)$$

$$W0 = 0.6002499789314202$$

$$W1 = 0.02117765478091216$$

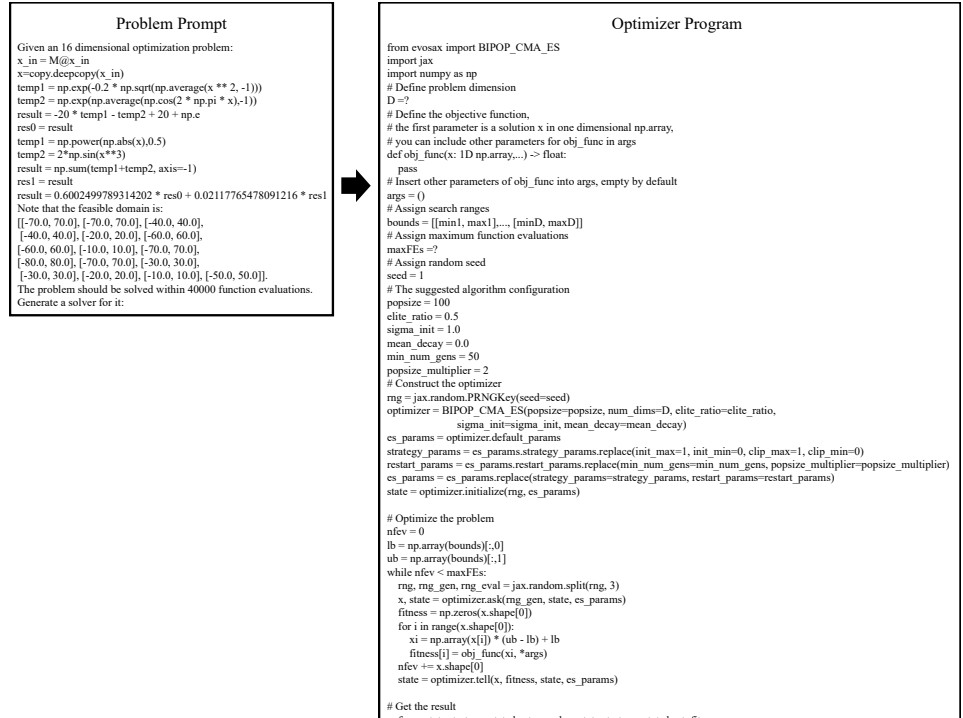

Figure 7: A unconstrained problem prompt (on the left, a Python version), and the optimization program (on the right) output by LLaMoCo-S. The corresponding 16-dimensional problem is constructed by a composition of two basic functions. Our LLaMoCo-S is prompted to output a competent optimizer for solving the problem within 40000 function evaluations, which in this case, is BIPOP-CMA-ES.

## F.2 SYNTHETIC CONSTRAINED EXAMPLE

We showcase the prompt and the generated optimization program (Figure 8) of a synthetic problem instance with some constraints, which has the following formulation:

$$Minimize: \quad f(x) = z_1^2 + 10^6 \sum_{i=2}^{D} z_i^2, z = x - o, X \in R^D, o \in R^D$$

$$s.t. :$$

$$h_0(x) : \sum_{i=1}^{D} \left( \sum_{j=1}^{i} y_j \right)^2 = 0, y = x - o$$

$$h_1(x) : \sum_{i=1}^{D-1} \left( y_i^2 - y_{i+1} \right)^2 = 0, y = x - o$$

Figure 8: A constrained problem prompt (on the left, a Python version), and the optimization program (on the right) output by LLaMoCo-S. The corresponding 23-dimensional problem is one of the basic functions, with two additional quality constraints. Our LLaMoCo-S is prompted to output a competent optimizer for solving that problem within 20000 function evaluations, which in this case, is SLSQP. We note that the GPT-4 Turbo attain the same answer on this problem. However, the configurations suggested by LLaMoCo-S achieve higher optimization performance against GPT-4 Turbo that adopts the default configurations.

F.3 REALISTIC EXAMPLE

We showcase the prompt and the generated optimization program (Figure 9) of a realistic problem instance with a large number of constraints yet with a relatively simpler objective function, which holds a different problem structure against the synthetic problems, which has the following formulation:

$$Minimize: \quad f(x) = 35x_1^{0.6} + 35x_2^{0.6}$$
$$s.t.:$$
$$h_1(x) : 200x_1x_4 - x_3 = 0$$
$$h_2(x) : 200x_2x_6 - x_5 = 0$$
$$h_3(x) : x_3 - 10000(x_7 - 100) = 0$$
$$h_4(x) : x_5 - 10000(300 - x_7) = 0$$
$$h_5(x) = x_3 - 10000(600 - x_8) = 0$$
$$h_6(x) = x_5 - 10000(900 - x_9) = 0$$
$$h_7(x)) = x_4 \ln(x_8 - 100) - x_4 \ln(600 - x_7) - x_8 + x_7 + 500 = 0$$
$$h_8(x) = x_6 \ln(x_9 - x_7) - x_6 \ln(600) - x_9 + x_7 + 600 = 0$$

**Problem Prompt**

```
Given an 9 dimensional optimization problem:
$\begin{aligned}
Minimize:\quad&f(x)=35x_{1}^{0.6}
+35x_{2}^{0.6} \\s.t.\quad &: \\&
h_1(x):200x_1x_4-x_3=0\\&
h_{2}(x):200x_{2}x_{6}-x_{5}=0\\&
h_{3}(x):x_{3}-10000(x_{7}-100)=0 \\&
h_4(x):x_5-10000(300-x_7)=0\\&
h_{5}(x)=x_{3}-10000(600-x_{8})=0 \\&
h_{6}(x)=x_{5}-10000(900-x_{9})=0 \\&
h_{7}(x))=x_4\ln(x_8-100)-x_4\ln(600-x_7)-x_8+x_7+500=0 \\&
h_{8}(x)=x_6\ln(x_9-x_7)-x_6\ln(600)-x_9+x_7+600=0
\end{aligned}$
Note that the feasible domain is:
[[0.0, 10.0], [0.0, 200.0], [0.0, 100.0],
[0.0, 200.0], [1000.0, 2000000.0], [0.0, 600.0],
[100.0, 600.0], [100.0, 600.0], [100.0, 900.0]].
The problem should be solved within 10000 function evaluations.
Generate a solver for it:
```

**Optimizer Program**

```python
from advanced_global_optimizers.constrained_solvers import GA_TDX
from advanced_global_optimizers import Problem
import numpy as np
# Define problem dimension
D =?
# Define the objective function,
# the first parameter is a solution x in one dimensional np.array,
# you can include other parameters for obj_func in args
def obj_func(x: 1D np.array,...) -> float:
    pass
# Insert other parameters of obj_func into args, empty by default
args = ()
# Define the constraints
def h_cons_1(x) -> float:
    pass
...

def g_cons_1(x) -> float:
    pass
...

constraints = [
    {'type': 'eq', 'fun': h_cons_1},
    ...
    {'type': 'ineq', 'fun': g_cons_1},
    ...
]
# Assign search ranges
bounds = [[min1, max1],..., [minD, maxD]]
# Assign maximum function evaluations
maxFEs =?
# The suggested algorithm configuration
param = {
    'beta': 0.1,
    'gamma': 3,
    'NP': 50,
}
# Construct the problem and optimizer
problem = Problem(D, obj_func, bounds, maxFEs, args, constraints)
optimizer = GA_TDX(param)
# Optimize the problem
result = optimizer.optimize(problem)
```

Figure 9: A realistic problem prompt (on the left, a LaTeX version), and the optimization program (on the right) output by LLaMoCo-S. The corresponding 9-dimensional problem holds an out-of-distribution structure, with far more constraints than the problem instances LLaMoCo-S has ever seen. Our LLaMoCo-S is prompted to output a competent optimizer for solving that problem within 10000 function evaluations, which in this case, is an advanced GA-TDX algorithm specialized in constraints handling. We note that the GPT-4 Turbo suggests a DE algorithm for this problem, which is hardly adopted for solving constrained problems.

