# OpenReview forum: "LLaMoCo: Instruction Tuning of Large Language Models for Optimization Code Generation"
_ICLR.cc/2025/Conference — Submitted to ICLR 2025_

### Official Review · Reviewer_eFRN · 2024-10-24

**Soundness:** 3
**Presentation:** 3
**Contribution:** 2
**Rating:** 5
**Confidence:** 3

**Summary:**

This paper introduces LLaMoCo, a new framework for fine-tuning LLMs to solve optimization problems. The contributions of this paper are two-fold: (1) a novel fine-tuning dataset and (2) a new training warm-up strategy for training leveraging contrastive learning. Experimental evaluations demonstrate that LLaMoCo's models perform well on their held-out test set and realistic problems.

**Strengths:**

1. The authors have developed and plan to release a novel dataset designed to teach language models to solve optimization problems. This represents a significant contribution to both researchers and practitioners.

2. The experimental results are compelling. I especially appreciate Table 3, where the proposed method strongly performs on realistic optimization problems (rather than toy problems).

**Weaknesses:**

1. This paper lacks novelty. This paper primarily focuses on fine-tuning OSS LLMs for a specific domain. The main approach is straightforward from this perspective: the authors adjusted prompts (specifically, framing problem descriptions in Python or Latex) and developed a new dataset. Could authors emphasize the unique technical challenges associated with this domain?

2. The contrastive warm-up technique in this paper seems out of place. This technique does not appear to be specifically tailored to optimization problems. Could it be beneficial for fine-tuning in other domains as well? If not, what are the reasons? I would suggest separating this novel technique into a dedicated paper or clarifying how it suits the domain under discussion. The ablation study in Figure 4 is not very convincing, as it was tested with only a single configuration, making the results dependent on that specific setup.

**Questions:**

1. Is the current dataset format truly optimal? For instance, could leveraging CoT enhance performance? Similarly, would implementing multi-turn iterative improvements for optimization code be a promising approach?

2. Could the proposed method be compared with previous non-LLM-based automatic algorithm selection approaches? Automatic algorithm selection for optimization problems is a well-established research area with a rich body of existing work.

3. Could the specific technical challenges unique to this optimization domain be highlighted? (see Weakness 1)

4. Would it be reasonable to separate the contrastive warm-up technique into a standalone paper or clarify that this technique is highly specialized for the domain under consideration? (see Weakness 2)

---

> ### Author Response · Authors · 2024-11-22
> **Response to Reviewer #eFRN (part 1/2)**
>
> We appreciate the reviewer for the valuable and constructive comments. We are grateful for your recognition of LLaMoCo as a significant contribution to the optimization domain, with compelling generalization performance across toy problems and realistic problems. Below, we provide point-by-point responses to address your remaining concerns.
>
> **[W1 & Q3, unique techinal challenges]**
>
> We would like to clarify that fine-tuning LLMs for specific domains, such as the optimization domain addressed in this paper, presents unique and domain-specific challenges. The primary contribution of LLaMoCo lies in identifying these challenges and proposing novel methodologies to overcome them. Below, we emphasize the unique technical challenges of adapting LLMs for the optimization domain, highlighting our contributions through three key aspects:
>
> Novelty 1: LLaMoCo is the first instruction-tuning framework for adapting general LLMs as an efficient and effective optimization tool. Existing works, such as OPRO [1], primarily rely on iterative prompting of LLMs to optimize solutions. However, these approaches suffer from a) unsatisfactory optimization performance due to the limited domain-specific knowledge of general LLMs. b) inefficient inference modes, which consume an extremely large number of tokens. In contrast, LLaMoCo directly injects optimization knowledge into LLMs through instruction tuning, representing a novel and systematic approach to leveraging LLMs for optimization tasks. The subsequent novelties address the unique challenges of this process and directly target the limitations (a) and (b) of existing works.
>
> Novelty 2: Representing and collecting the optimization knowledge (Section 3.1). We propose a stipulated and unique code-to-code description format to represent optimization problems and their corresponding effective optimizers. This universal representation format facilitates automated data collection and simplifies the adaptation of LLMs to the optimization domain. To collect sufficient optimization knowledge,  we have proposed a novel optimization problem generation process which is capable of synthesizing a large number of diverse optimization problem instances, which are further utilized by our proposed automated benchmarking procedure to obtain the most effective optimizer codes—the optimization knowledge.
>
> Novelty 3: Enhancing the training effectiveness (Section 3.2). We explicitly inject the obtained optimization knowledge into general LLMs through efficient instruction tuning. The unique challenges in this process are a) code similarity understanding issues (lines 262-269) and b) data imbalance issues (lines 299-301). We additionally introduce a contrastive warmup for aligning the code similarity, and an example-proportional mixing strategy to re-balance the training data, both of which enhances the training efficiency and stability.
>
> Besides, we have to note that the contribution of LLaMoCo is not limited to the above novel proposals. The dataset construction, the fine-tuning strategy and all observed empirical results would provide in-depth insights and a profound impact on the future development of a combination of LLMs and optimization domain. Following your valuable suggestion, we have refined the above discussion into the related works section in the revised paper to highlight these novelties (lines 125-128, 142-145, 169-173, 181-187, colorred in blue).
>
> [1] Chengrun Yang, et al. "Large language models as optimizers.” arXiv preprint arXiv:2309.03409, 2023.
>
> **[Q2, compare with algorithm selection methods]**
>
> We argue that comparing LLaMoCo with algorithm selection methods may not be entirely appropriate due to two key distinctions:
>
> 1. **Code Generalization Advantage**: Unlike algorithm selection methods, which merely choose the best algorithm from a predefined pool, LLaMoCo generates complete optimizer source codes. These codes not only specify the selected algorithm but also include the necessary implementation details, ensuring compatibility with various optimization problem descriptions. This level of generalization is crucial for real-world applications, where optimization problems often require customized code beyond the scope of standard algorithm selection.
> 2. **Hyperparameter Tuning**: LLaMoCo also performs hyperparameter tuning as part of the optimization code generation process. By tailoring hyperparameter values to the specific problem instance, LLaMoCo provides a level of configurability that algorithm selection methods cannot achieve.
>
> Following your suggestion, we have added some discussion in Section 4.1, lines 349-354 of the revised paper (colorred in blue), where we highlight the above technical differences between LLaMoCo and Algorithm Selection methods. We kindly request the reviewer to check for the updates.

---

> ### Author Response · Authors · 2024-11-22
> **Response to Reviewer #eFRN (part 2/2)**
>
> **[W2 & Q4, contrastive warm-up]**
>
> We are honoured that the reviewer finds the contrastive warm-up component noteworthy enough to merit a standalone paper. However, we clarify that it is an integral part of LLaMoCo, tailored specifically for the optimization domain. In Section 3.2 (lines 262-269), we discuss two domain-specific challenges that necessitate the contrastive warm-up to facilitate the subsequent SFT.
>
> 1. In the optimization domain, the number of optimization problems is far more than the number of optimizers. Hence, multiple problems can share the same effective optimizer despite differing descriptions. Without a contrastive warm-up to align the hidden embeddings, the model struggles to generalize well across these cases.
> 2. Minor changes in the problem descriptions could significantly alter the optimization properties, resulting in different effective optimizers. The contrastive warm-up pulls the hidden embeddings of these similar but distinct problems apart, enhancing the model’s ability to distinguish them.
>
> While we admit the idea of contrastive warm-up could potentially be applied to other domains with similar alignment issues, this should not be a criticism point of our paper. On the contrary, this generalizability reflects the novelty and broad impact of our proposal! Note: we have added the above discussion into the revised paper, Section 5, lines 533-538 (colorred in blue), as a promising future direction for potential practitioners.
>
> Besides, we understand your concern that we only showcase the effectiveness of the contrastive warm-up of LLaMoCo-S on $\mathbb{I}_{eval}$ in the ablation studies (Figure 3). We have expanded these to include performance gain curves for eight more scenarios (three models—LLaMoCo-S/M/L—and three test sets from Table 1).  These new results, presented in Figure 6, Appendix E.5 of the revised paper (colorred in blue),  consistently demonstrate the learning enhancement through the contrastive warm-up. We kindly request the reviewer to examine these figures.
>
> **[Q1, leverage CoT & multi-turn code optimization conversations]**
>
> We clarify that under the motivation and vision of LLaMoCo, the current dataset format seems to be optimal. To be specific, the core motivation of LLaMoCo is to provide an alternative way of leveraging LLMs for the optimization domain which could achieve superior optimization performance with minimal computational resources. To this end, we design the dataset format as the prompt-answer pair, which facilitates both end-to-end training and inference. Once trained, LLaMoCo generates the desired optimization codes in a single conversation turn, directly supporting this motivation.
>
> While we acknowledge that constructing a CoT-based training dataset could potentially enhance LLaMoCo's generalization through reasoning dynamics, this approach diverges from our primary goals and requires significantly more human efforts and costs: a) it would significantly increase token consumption during inference, and b) constructing such a dataset would require substantially more domain expertise, whereas our current dataset is fully automated, generated through benchmarking and grid search. We hope this explanation clarifies our design choice. Besides, while we acknowledge that leveraging multi-turn code optimization might potentially enhance the performance of the generated optimizers, it comes with notable drawbacks. First, it requires users to provide additional feedback and guidance to the LLMs, which demands a certain level of expertise. Second, multi-turn code optimization consumes significantly more computational resources (e.g., token usage), making it far less efficient and economical compared to LLaMoCo's single-turn approach.
>
> We thank the reviewer for such valuable comments. We have added the core part of the above discussion into the revised paper, Section 5, lines 533-538 (colorred in blue), as promising directions for further development.

---

> > ### Comment · Reviewer_eFRN · 2024-11-25
> >
> > I appreciate the authors' detailed responses. However, I remain unconvinced that the paper adequately demonstrates "unique technical challenges" specific to the optimization domain. The presented novelties appear to be applications of existing approaches (instruction tuning, dataset construction, and contrastive learning) rather than truly domain-specific technical innovations. Moreover, while the authors argue that their contrastive warm-up technique is specifically tailored to optimization problems, the justification for this claim remains weak. Therefore, I maintain my original rating of 5.

---

> > > ### Author Response · Authors · 2024-11-25
> > > **Further Response (part 1/2)**
> > >
> > > We appreciate for your timely feedback. We are very happy that  we have addressed half of your concerns (Q1 and Q2). Let us further explain and address your remaining concerns in Q3 and Q4.
> > >
> > > First of all, we would like to highlight that the primary contribution of our paper lies in the design of a novel framework that accepts optimization problem descriptions at the Latex or Python code level and enables LLMs to direcctly generate codes for solving these problems. Developing this framework represented numerous challenges, which are not trivial—they involve fundamental issues in representing and interpreting optimization problems that LLMs can process effectively, as well as ensuring the generated solutions are accurate and efficient. To meet the unique requirements of this framework, we developed tailored adaptations and extensions of existing techniques  (instruction tuning, dataset construction, and contrastive learning). We respectfully do not fully understand the reviewer’s comment regarding the "truly domain-specific technical innovations". The core challenge of our work lies in constructing the framework and identifying the fundamental problems within it. In such a context, reinventing the wheel in terms of technical methods would detract our focus from solving these high-level challenges effectively.
> > >
> > > **[Unique challenges in LLaMoCo]**
> > >
> > > Below, we summarize each of the unique challenges encountered in developing LLaMoCo and the novel design adaption/extension of existing techniques we incorporated in a point-to-point way:
> > >
> > > 1. Unique dataset construction.
> > > a) Challenge 1: **How to represent optimization problem in language description?** To address this, we proposed a stipulated problem formulation structure which facilitates the subsequent automated problem collection.
> > >
> > >     b) Challenge 2: **How to efficiently attain sufficient optimization problem instances to facilitate training of LLMs?** To address this, we proposed a novel and **fully automated problem synthesizing procedure (in Section 3.1, 197-221)**, which not only eases the random combination of objective functions, but also provides diverse optional constraints.
> > >
> > >     c) Challenge 3: **How to gather high-quality optimization knowledge for the training problem instances?** To address this, we have carefully prepared a high-performance algorithm pool and then proposed a **large-scale grid search-based benchmarking procedure** **(in Section 3.1, 222-240, and Appendix A)** to attain the most effective optimization code for each problem instance automatically.
> > >
> > > 2. Unique instruction tuning process.
> > >
> > >     a) Challenge 1: **How to deal with the specific semantic alignment issue raised in optimization domain?** As we mentioned in our previous rebuttal (W2 & Q4, a. b.), in optimization domain, there are two special phenomenons: First, some totally different problems might share the same effective optimizer. Second, when we present the optimization problem in python and latex code, a slight code modification would lead to different target optimizer. These two points raises unique challenge if we want to learn an effective code-to-code model in LLaMoCo. To address this, we introduced **contrastive warm-up (in Section 3.2, lines 262-298)** to allign the code-level semantics before the normal instruction tuning (SFT).
> > >
> > >     b) Challenge 2: **How to deal with the data imbalance issue raised in optimization domain?** In optimization domain, a small number of  optimizers might dominate optimization performance on a large number of problem instances, causing the imbalance in our training data. To address this, we propose using the **example-proportional mixing strategy (in Section 3.2, lines 299-312)** to re-balance the data distribution for training stability.
> > >
> > > 3. Unique evaluation procedure.
> > >
> > >     a) Challenge 1: **How to measure the performance of fine-tuned models in a comprehensive way?** To address this, we have designed the **four performance metrics (in Section 4.1, lines 355-365, and Appendix D)**: Code Error Rate (Err.), Code Recovery Cost (Rec.), Optimization Performance (Perf.), and Computational Overhead (Comp.) to evaluate all aspects of LLaMoCo.
> > >
> > >     b) Challenge 2: **How to systematically analyse the capability of fine-tuned models?** To address this, we investigate the **scaling law (in Section 4.2, lines 405-411, and Appendix E.3)** of our LLaMoCo on different model sizes and training dateset sizes, as well as the **zero-shot performance** **(in Section 4.2, lines 412-446, and Appendix E.2)** on totally unseen realistic problems.

---

> > > ### Author Response · Authors · 2024-11-25
> > > **Further Response (part 2/2)**
> > >
> > > In a word, LLaMoCo indeed represents a novel sub-field of LLM for Optimization. The in-depth experimental observations (those already presented in the original version and those we have added according to all reviewers’ constructive suggestions) fully demonstrate the correctness and novelty of our special design efforts. The dataset (its format and construction), the training paradigm (contrastive warm-up + SFT) and the in/out-of-distribution evaluation procedure (four comprehensive performance metrics and each analysis module) all contribute to a systematic and easy-to-follow guideline for future practitioners, hence showing broad impact.  **We really hope the reviewer could understand that to develop a comprehensive framework, LLaMoCo is unavoidably built upon a stack of advanced technologies, however, the efforts and insights behind these technical usage to make LLaMoCo efficient and effective are more important.**
> > >
> > > **[Contrastive warm-up]**
> > >
> > > Given the further explanation above, we hope the reviewer could understand **the contrastive warm-up is exactly part of our chain-of-thoughts** to address the ultimate goal of LLaMoCo: generate effective optimizer for optimization problem at code-level. Such an interesting finding could benefit for other domain and definitely deserves further investigation (as you suggested). As we mentioned, we have highlighted in the added future work part in Section 5, lines 533-535 of the revised paper to notify future readers.
> > >
> > > At last, we hope the above further clarifications could address your remaining concerns. We sincerely hope you will revisit your rating in light of this additional discussion. Thank you for your time and effort in reviewing our responses.

---

> > > ### Author Response · Authors · 2024-11-26
> > > **Request for further feedback**
> > >
> > > Dear reviewer #eFRN, since the discussion period is extended, we respectifully request you for futher feedback on our response to youe newly posted comments. Is there specific experimental analysis we should conduct to further address your concerns? We are open to any suggestions from you. Thanks for your precious time!

---

> > > > ### Author Response · Authors · 2024-12-02
> > > >
> > > > Dear Reviewer #eFRN:
> > > >
> > > > The discussion period will end soon. If you still have any concerns, we look forward to hearing from you and will address them before the discussion ends.
> > > >
> > > > Best regards, the authors

---

### Official Review · Reviewer_ugM1 · 2024-10-27

**Soundness:** 3
**Presentation:** 2
**Contribution:** 3
**Rating:** 6
**Confidence:** 4

**Summary:**

This paper presents LLaMoCo, a pioneering framework that maps optimization problem descriptions directly to expert-level optimization code through instruction tuning. By creating a comprehensive dataset of (problem, best-solver) pairs and using a two-phase training strategy, even a small model (350M parameters) can surpass GPT-4 in selecting and generating appropriate optimizers for both synthetic and realistic optimization tasks.

**Strengths:**

1. 350M parameter model achieves 81.8% optimization performance vs GPT-4's 74.2% (without prompting) while using only 2.4K tokens vs 3.5K tokens.
2. Data pipeline converts 6000 problems to 32570 training pairs through systematic benchmarking of 23 optimizers across different configurations.

**Weaknesses:**

1. Zero-shot evaluation tested on only 8 realistic problems, requiring more cases to validate the claims.
2. GPT-4 baseline with vector search not evaluated.
3. Grid search necessity on original problems is subtle, some parameters are hard to set without careful data observation, requiring further validation of selection appropriateness.

**Questions:**

1. Further experiments needed to demonstrate the combined effect of SFT and alignment.
2. Grid search "best" performance criteria not clearly defined, benchmarking process lacks clear evaluation metrics for optimizer selection.

---

> ### Author Response · Authors · 2024-11-22
> **Response to Reviewer #ugM1 (part 1/2)**
>
> We appreciate the reviewer for the valuable comments. Thank you for recognizing our work as pioneering, the superior performance of LLaMoCo compared to the GPT-4 model in solving optimization problems, and the systematic construction of our training dataset. To address your remaining concerns, we provide the following point-by-point responses.
>
> **[W1, add zero-shot evaluation]**
>
> We understand the reviewer's concern regarding the importance of testing zero-shot evaluation on a large dataset. However, we believe there may be a misunderstanding. To clarify, the first six problems in Table 3 represent individual engineering problem instances, while the latter two—HPO-B and Protein-Docking—are extensive problem collections containing hundreds of instances. For example, the Protein-Docking collection consists of diverse protein-protein complexes with varying structures, each presenting a challenging optimization landscape. In total, the number of tested problem instances amounts to 6 + 128 + 128 = 262 rather than 8.
>
> To further address your concern, we conducted additional testing of our trained model on a new realistic problem collection derived from the first six problems in Table 3. This collection, proposed by Kumar et al. [1], consists of 57 real-world constrained optimization problems sourced from a diverse range of engineering scenarios. The comparison results (averaged across all 57 problems) between our LLaMoCo and other baselines are presented in the following table:
>
> |  | OPRO | LMEA | CodeGen-Mono350M | Phi-2-2.7B | DeepSeekMathInstruct-7B | GPT-4 Turbo | Code Llama-7B | Llama2-70B | LLaMoCo-S | LLaMoCo-M | LLaMoCo-L |
> | --- | --- | --- | --- | --- | --- | --- | --- | --- | --- | --- | --- |
> | Err. | - | - | 99.487% | 98.131% | 68.921% | 40.148% | 99.344% | 99.473% | 5.984% | 5.479% | **5.359%** |
> | Rec. | - | - | 81.166% | 58.546% | 15.470% | 16.791% | 58.101% | 59.189% | 10.648% | 10.486% | **10.198%** |
> | Perf. | 31.832% | 27.549% | 26.477% | 33.460% | 58.044% | 61.468% | 49.481% | 47.146% | 84.462% | 87.279% | **88.135%** |
> | Comp. | 253k | 298k | 2.1k | 2.1k | 2.1k | 3.6k | 2.0k | 1.9k | 2.6k | 2.6k | 2.6k |
>
> The results further validate the effectiveness and superior performance of our LLaMoCo. We have added corresponding text content in lines 431-433 and the above results and discussion in Appendix E.2.
>
> [1] Kumar, Abhishek, et al. "A test-suite of non-convex constrained optimization problems from the real-world and some baseline results." *Swarm and Evolutionary Computation* 56 (2020): 100693.
>
> **[W2, add GPT-4 vector search baseline]**
>
> We have conducted a baseline experiment based on our understanding of “GPT-4 vector search.” Specifically, we utilized the GPT-4 vector embedding system to generate vectorized embeddings for all prompts in our training dataset. During testing, the tested prompt was also processed through the GPT-4 vector embedding system to obtain its vectorized embedding. We then identified the most similar prompt in the training dataset by calculating the L-2 distance between the embeddings. Finally, the tested prompt, along with the most similar prompt and its corresponding answer from the training dataset, were fed into the GPT-4 model to generate an optimizer code. In this setup, the most similar prompt and its corresponding answer serve as an example of in-context learning. We present the comparison results of this GPT-4 vector search baseline, GPT-4 baseline and our LLaMoCo-L on the test set $\mathbb{I}_{eval}$ in the following table:
>
> | Baseline | GPT-4 Turbo | GPT-4 vector search | LLaMoCo-L |
> | --- | --- | --- | --- |
> | Err. | 41.667% | 9.336% | **5.509%** |
> | Rec. | 13.072% | 12.853% | **10.461%** |
> | Perf. | 74.248% | 79.944% | **83.451%** |
> | Comp. | 3.5k | 7.1k | **2.4k** |
>
> Two key observations can be made: a) Providing GPT-4 with an example prompt-answer pair similar to the tested prompt significantly reduces the error rate of the generated optimizer code.  b) However, this prompting strategy consumes twice as many tokens as directly prompting GPT-4, making it inefficient—especially when compared to LLaMoCo, which requires only 2.4k tokens to achieve superior optimization performance. This highlights the importance of LLaMoCo in efficiently adapting LLMs to solve optimization problems. We have added the above results and discussion in Section 4.4, Table 4 of the revised paper (colorred in blue).

---

> ### Author Response · Authors · 2024-11-22
> **Response to Reviewer #ugM1 (part 2/2)**
>
> **[W3 & Q2, grid search criteria and correctness]**
>
> 1. grid search criteria
>
>     We kindly refer the reviewer to Section 3.1, lines 234-240 and Appendix A.2 for all details of our benchmarking criteria. In summary, we have selected 23 representative optimizer commonly used to solve different optimization problems and then applied a two-step procedure to identify the most effective optimizer for each problem instance in our instruction tuning set. Step 1: according to the configuration space we provided in Table 6 of the Appendix, we identify the best-performing configurations of the 23 optimizers for the given problem instances. Step 2: The optimizer that achieves the highest optimization performance by using the found best-performing configuration is labelled as the most effective optimizer for that problem instance.
>
> 2. grid search correctness
>
>     Following your suggestions about the grid search granularity we chose (as shown in Table 6 of the Appendix). We conduct two additional benchmarking processes, with half and double granularity of our original setting. For example, if a hyper-parameter holds four optional values in our setting: [0.2, 0.4, 0.6, 0.8], half granularity denotes [0.2, 0.8], double granularity denotes [0.1, 0.2, 0.3, 0.4, 0.5, 0.6, 0.7, 0.8], etc. We present the averaged optimization performance of the most effective optimizers on our problem set searched by these two granularities, normalized by our original granularity’s performance, as well as the averaged searching wall time for one problem instance in the following table:
>
>     |  | half | our setting | double |
>     | --- | --- | --- | --- |
>     | performance | 71.793% | 1 | 102.344% |
>     | wall time | 6s | 211s | 6379s |
>
>     The results reveal an evident tradeoff between the searching effectiveness and the searching efficiency of different grid search granularities. The searching wall time increases exponentially since there are 4-5 hyper-parameters in an optimizer. However, the performance improvement obtained by spending so much computational resources is only 2.344%.  We believe this result could validate the selection appropriateness of our grid search granularity. We have added the above discussion in Appendix E.4 of the revised paper (colorred in blue). We also have added some text content in lines 239-240 (colorred in blue) of the revised paper to guide readers for this discussion.
>
>
> **[Q1, combined effect of SFT and contrastive alignment]**
>
> We understand your concern since we only showcase the performance gain curve of our LLaMoCo-S on the left of Figure 3. We have added the same performance gain curves of the other eight scenarios (three models LLaMoCo-S/M/L and three test sets in Table 1). These curves are now presented in Figure 6, Appendix E.5 of the revised paper, where we can observe consistent learning enhancement by introducing our proposed contrastive learning warmup. We kindly request the reviewer to examine these figures. We have also added some text content in lines 475-476 of the revised paper to guide readers to review these results.

---

> ### Author Response · Authors · 2024-11-28
> **Request for further feedback**
>
> Dear reviewer #ugM1:
>
> Since the discussion period has been extended, we respectifully request your feedback on our responses. In these responses, we have conducted additional experiments and provided in-depth discussions to address your concerns. If there are any further concerns, we are eager to continue this discussion and address them. We look forward to hearing from you.
>
> Best regards,
> the authors

---

> > ### Comment · Reviewer_ugM1 · 2024-11-29
> >
> > Thank you for your response. While part of my question has been addressed, I still have several concerns, three of which are as follows:
> >
> > 1. In my understanding, code generated by a 350M parameter model is typically quite poor. There needs to be sufficient examples demonstrating its effectiveness and executability, as well as analysis of why it fails.
> > 2. The improvement from 350M to 7B parameters appears relatively modest. What accounts for this? I'm particularly puzzled about what exactly the model is learning - is it merely learning the code format for problem-solving?
> > 3. Could you please provide a breakdown of the frequency of each optimizer's occurrence in training and inference respectively?

---

> > > ### Author Response · Authors · 2024-11-30
> > > **Response to your futher feedback (part 1/2)**
> > >
> > > We appreciate the reviewer for the timely feedback. We provide following point-to-point responses to address your remaining concerns.
> > >
> > > **[Q2.1, performance difference]**
> > >
> > > We would clarify that the seemly “modest” improvement is caused by the  **normalized performance metric** (detailed in Appendix D), which scales optimization performance by the objective value range of the target problem instance. These ranges can be extremely large (e.g., $10^{20}$), causing performance differences to appear smaller than they are. For instance, in Table 1, the small model (LLaMoCo-S, 350M) achieves a performance of 81.843%, which seems "similar" to the medium model (LLaMoCo-M, 2.7B) at 83.369%. However, this 1.5% difference reflects a significant performance gap in absolute terms. The reason we normalize the optimization performance is that given various objective value scales in different problem instances, it is unreasonable to compute the absolute average final objective value across problem instances. Additionally, we believe that such normalization improves the table readability of the results.
> > >
> > > **[Q2.2, what does LLaMoCo learn?]**
> > >
> > > We would clarify that LLaMoCo is not merely learning the code fomat for problem-solving. Instead, it learns a comprehensive model that generates effective and executable optimizer code for solving optimization problem. We explain this from four aspects:
> > >
> > > 1. **LLaMoCo learns how to understand an optimization problem**. We train LLaMoCo to generate optimization code in an end-to-end style based on the stipulated problem description format. By doing this, LLaMoCo learns to understand the language description of the given optimization problem including the objective, searching range, number of function evaluations, number of dimensions and additional contraints during its training.
> > > 2. **LLaMoCo learns how to select a suitable optimizer for the given problem**. Thanks to the end-to-end instruction set we have constructed, LLaMoCo is capable of locating the most effective optimizer for the given optimization problem according to its understanding of that problem. Unlike algorithm selection methods, which merely choose the best algorithm from a predefined pool, LLaMoCo generates complete optimizer source codes. These codes not only specify the selected algorithm but also include the necessary implementation details, ensuring compatibility with various optimization problem descriptions.
> > > 3. **LLaMoCo learns how to configure the optimizer properly**. In specific, LLaMoCo performs hyperparameter tuning as part of the optimization code generation process. This originates from the fine-grained grid search-based benchmarking when we construct the instruction-tuning set (Section 3.1, lines 233-240). By tailoring hyperparameter values to the specific problem instance, we train LLaMoCo to provide a level of configurability according to its understanding about the problem isntance.
> > > 4. **LLaMoCo learns how to address the unique semantic alignment issue in optimization domain**. In LLaMoCo, when we represent an optimization problem in a stipulated language format, the difference between two significantly different problems might be narrowed down (as we described in Section 3.2, lines 263-270). This motivates us to train LLaMoCo by the proposed contrasitive warmup first to align the semantic differences between diverse problem instances, hence improving the learning effectiveness of the subsequent SFT process.
> > >
> > > **[Q1, 350M model’s performance]**
> > >
> > > We would argue that according to the results we presented in Table 1, the optimization codes generated by our LLaMoCo-S (350M CodeGen model) show competitive error rate compared with larger LLaMoCo models and signicantly superior error rate to the general LLMs baselines we have compared, which demonstrates the consistent training effectiveness of LLaMoCo on foundation models with different capacities. To demonstrate LLaMoCo’s effectiveness and executability, we provided in the original paper:
> > >
> > > 1. some generation examples of LLaMoCo-S (350M) in Appendix F.1, F.2 and F.3 , where this small model is capable of generating fully executable and effective optimizer programs for unconstrained problems, constrained problems and realistic problems. We respectifully request the reviewer to check for these demonstrations.
> > > 2. an anonymous tutorial project accessible online (https://anonymous.4open.science/r/LLaMoCo-5125), where we also provide this 350M model and step-by-step generation codes for further validation. We respectifully request the reviewer to run the “LLaMoCo_For_Review.ipynb” file to validate the robust effectiveness and executability of LLaMoCo-S. You can replace the showcase problem we used there by your optimization problem following the stipulated format there and observe the consistent correctness of the generated code.

---

> > > > ### Comment · Reviewer_ugM1 · 2024-11-30
> > > >
> > > > Appendix D doesn't seem to provide an exact formula for the performance, but rather states each metric separately. So does it have an exact formula? (This was also my original Q2)

---

> > > ### Author Response · Authors · 2024-11-30
> > > **Response to your futher feedback (part 2/2)**
> > >
> > > **[Q3, optimizer’s occurrence frequency]**
> > >
> > > We provide following two tables that present the frequency of the 23 optimizers in our proposed advanced optimizer pool.
> > >
> > > The frequency in training set.
> > >
> > > | **SAMR-GA** | **GA-TDX** | **Vanilla DE** | **DEAP-DE** | **HECO-DE** | **MadDE** | **AMCDE** | **Vanilla PSO** |
> > > | --- | --- | --- | --- | --- | --- | --- | --- |
> > > | 0.02% | 17.8% | 20.0% | 0.03% | 0.03% | 1.67% | 0.03% | 0.03% |
> > > | **GLPSO** | **sDMS-PSO** | **DTPSO** | **SEP-CMA-ES** | **BIPOP-CMA-ES** | **MMES** | **Vanilla BO** | **LA-MCTS** |
> > > | 2.78% | 0.80% | 4.40% | 0.02% | 13.5% | 0.02% | 0.02% | 0.05% |
> > > | **SA** | **Dual Annealing** | **NSA** | **SLSQP** | **Trust-Constr** | **COBYLA** | **L-BFGS-B** |  |
> > > | 0.10% | 10.8% | 0.17% | 24.6% | 0.03% | 3.08% | 0.02% |  |
> > >
> > > The frequency in testing set.
> > >
> > > | **SAMR-GA** | **GA-TDX** | **Vanilla DE** | **DEAP-DE** | **HECO-DE** | **MadDE** | **AMCDE** | **Vanilla PSO** |
> > > | --- | --- | --- | --- | --- | --- | --- | --- |
> > > | 0.03% | 16.7% | 18.9% | 0.03% | 0.05% | 1.60% | 0.05% | 0.02% |
> > > | **GLPSO** | **sDMS-PSO** | **DTPSO** | **SEP-CMA-ES** | **BIPOP-CMA-ES** | **MMES** | **Vanilla BO** | **LA-MCTS** |
> > > | 2.55% | 1.05% | 4.72% | 0.05% | 14.6% | 0.02% | 0.03% | 0.02% |
> > > | **SA** | **Dual Annealing** | **NSA** | **SLSQP** | **Trust-Constr** | **COBYLA** | **L-BFGS-B** |  |
> > > | 0.13% | 10.6% | 0.22% | 23.3% | 0.05% | 5.20% | 0.02% |  |
> > >
> > > From the tables, we can observe that there is a data imbalance challenge (e.g., 20.4 % is Vanilla DE optimizer) considering instruction-tuning general LLMs for solving optimization problems in LLaMoCo, which motivates us to employ a balanced data sampling strategy (Section 3.2, lines 299-312) to aviod biased training. This further highlights the importance of LLaMoCo since it aims to generate most effective optimizer code not only for the majority problems that can be easily solved by popular optimizer such as GA, DE, CMA-ES and SLSQP, but also for uncommon problems which require specialized optimizer. Such optimization knowledge, as we discussed in Section 4.4, makes LLaMoCo not only more effective than general LLMs such as GPT-4 (it outputs DE or SLSQP for almost all tested problems), but also more friendly for users since when a user (with limited optimization knowledge) has an optimization problem to solve, LLaMoCo could automatically generate a specialized optimizer in an end-to-end style. In contrast, if this user prompts GPT-4, the answer is likely a DE optimizer, which might lead to poor performance for his or her optimization problem.

---

> ### Author Response · Authors · 2024-11-30
>
> There are $N_p = 2570$ problem instances in the testing set $\mathbb{I}_{eval}$ (We have constructed 32570 problems, 30000 of them are used for the instruction tuning, the rest are used for testing). For each problem $p_i$ in the testing set, we feed its prompt to LLaMoCo and obtain the generated optimizer program. We then run the program to optimize $p_i$. There are three cases to compute the performance of LLaMoCo on $p_i$, which we denote as $Perf_i$:
>
> 1. If the generated optimizer code is executable with no errors, we run the program to optimize $p_i$ for 5 independent runs. Then the performance is calculated as $Perf_i = \frac{1}{5} \sum_{j=1}^{5} \frac{f_{i,j}^* - f_i^*}{f_{i,j}^0 - f_i^*}$. Where $f_{i,j}^0$ is the initial objective value in j-th run, $f_{i,j}^*$ is the best objective value found by the generated optimizer code, $f_i^*$ is an approximation of the optimum of $p_i$ which we obtain from the large scale benchmarking.
> 2. If the errors can not be resolved within one turn of debug conversation, we set $Perf_i$ as 0.
> 3. If there are runtime errors, we prompt LLMs to debug this program. If the errors are resolved within one turn of debug conversation, we calculate a recovery cost $r_i = \frac{L_{err}^i}{L^i}$ as the proportion of lines in the generated codes that need to be revised, where $L_{err}$ denotes the number of error lines, $L$ denotes the total number of  lines. We then run the revised program to optimize $p_i$ for 5 independent runs and calculated the performance as $Perf_i = (1-r_i) \times \frac{1}{5} \sum_{j=1}^{5} \frac{f_{i,j}^* - f_i^*}{f_{i,j}^0 - f_i^*}$ . We give a punishment term to the normalized performance to reflect the errors raised in the generated optimizer code.
>
> We provide the exact formula of this performance metric as below:
>
> $$
> Perf_i =  \begin{cases}\frac{1}{5} \sum_{j=1}^{5} \frac{f_{i,j}^* - f_i^*}{f_{i,j}^0 - f_i^*} & \text{Case 1;}\\\0 & \text{Case 2;}\\\\(1-r_{i}) \times \frac{1}{5} \sum_{j=1}^{5}\frac{f_{i,j}^* - f_i^*}{f_{i,j}^0 - f_i^*} & \text{Case 3.}\end{cases}
> $$
>
> At last, the performance of LLaMoCo on testing set is the average across all problem instances:
>
> $$
> Perf = \frac{1}{N_p} \sum_{i =1}^{N_p} Perf_i
> $$
>
> We thank the reviewer for this valuable comments, we will update the Appendix D to make the metric calculation more clear for our readers.

---

> > ### Comment · Reviewer_ugM1 · 2024-11-30
> >
> > Thank you. But what I mean is: "exact formula", such as how the Comp. term is calculated, rather than an ambiguous description of "average number of tokens (input+output)", which is obviously unlikely to be used directly in calculations.
> >
> > Without an exact formula, the effectiveness of the paper will be greatly reduced.

---

> > > ### Author Response · Authors · 2024-11-30
> > >
> > > In LLaMoCo, we calculate the Comp. term of our fine-tuned model and the baselines by counting the tokens consumed by a LLM to generate a complete optimizer code for the given problem prompt. In specific, we first use "transformers.AutoTokenizer" interface to initialize a tokenizer. Then for each problem instance $p_i$ in the testing set, we use the tokenizer to tokenize the problem prompt of $p_i$ and record the number of tokens in the token list as $N_{in}^i$. Once the LLM generates a complete optimizer code, we use the same tokenizer to tokenize the generated code string and record the number of tokens in the token list as $N_{out}^i$. Then the Comp. term is calculated as $Comp. = \frac{1}{N_p} \sum_{i=1}^{N_p}(N_{in}^i + N_{out}^i)$. We hope this elaboration could clear your concern.
> > >
> > > Thank you for the precious time and valuable suggestion! We will update this exact formula for the Comp. term, as well as the above Perf. term into the revised paper.

---

> > > > ### Author Response · Authors · 2024-12-02
> > > >
> > > > Dear Reviewer #ugM1:
> > > >
> > > > Since the discussion period will end soon, we respectifully request for your further feedback. If you still have any concerns, we look forward to hearing from you and address them before the discussion ends.
> > > >
> > > > Best regards, the authors

---

### Official Review · Reviewer_1QNj · 2024-10-27

**Soundness:** 3
**Presentation:** 2
**Contribution:** 3
**Rating:** 6
**Confidence:** 3

**Summary:**

This paper proposes a data generation and instruction tuning method for optimization-problem-solving LLMs. The authors conduct comprehensive experiments to demonstrate the optimization capabilities of the instruction-tuned LLMs and analyze the contribution of each component of the method.

**Strengths:**

1. This paper introduces the first complete framework for training LLMs to solve optimization problems, including instruction-tuning dataset construction and detailed methods for training. The method is well-described and effective, making a significant contribution to the optimization community.

2. The experiments demonstrate performance improvements on both synthetic and realistic problem sets. across different scales of LLMs, highlighting the generalization and effectiveness of LLaMoCo.

**Weaknesses:**

1. Lack of sufficient novelty. Several key components of the method follow prior work [1-3], particularly the instruction-tuning approach (Section 3.2), which reduces its originality. Although this paper introduces the first instruction-tuning framework for optimization tasks, it primarily applies standard training techniques. The authors should emphasize their main innovations more clearly in the paper.

2. Writing. Figure 1 does not effectively highlight the main differences between LLaMoCo and previous methods, which is overly simplified. The authors should include more details of the method. There are typos in the caption of Figure 2 (wither -> either). The capitalization of “LaTeX” in the full paper is inconsistent.

[1]Problem definitions and evaluation criteria for the cec 2021 on single objective bound constrained numerical optimization.

[2]Unixcoder: Unified cross-modal pre-training for code representation.

[3]Exploring the limits of transfer learning with a unified text-to-text transformer.

**Questions:**

1. What is the impact of dataset size on the training performance? Will the performance of the models continue to improve when using more data?

2. How to control the quality of the synthesized tasks? Can we ensure that unsolvable optimization problems or problems with only trivial solutions are not synthesized?

3. Why does the computational overhead of the trained models increase? (in Table 1)

---

> ### Author Response · Authors · 2024-11-22
> **Response to Reviewer #1QNj (part 1/2)**
>
> We appreciate the reviewer for acknowledging our LLaMoCo as the first complete framework making a significant contribution to the optimization community. We are pleased that the reviewer finds our paper well-described and effective, with solid experimental results demonstrating both generalization and effectiveness. We hope the following point-by-point responses address the remaining concerns.
>
> **[W1, novelty]**
>
> We appreciate the reviewer’s concern and would like to clarify the unique innovations in LLaMoCo that set it apart from the referenced works [1], [2], and [3]:
>
> 1. We emphasize that while the dataset construction incorporates the synthetic process outlined in [1], we significantly augment this process by carefully introducing random constraints collected from extensive convex optimization literature (as detailed in lines 200–202 and 213-215). This data augmentation ensures that the final dataset includes not only the unconstrained problems from [1] but also a diverse and novel set of constrained optimization problems. This enhancement plays a critical role in improving the generalization performance of LLaMoCo. It also provides valuable insights and inspiration for designing effective training datasets to fine-tune LLMs for potential other optimization tasks in future work.
> 2. we would clarify the difference between the contrastive learning introduced in LLaMoCo and the approach used in Unixcoder [2]. The contrastive learning in Unixcoder involves two components: a) aligning representations of different modalities by aligning the hidden dropout masks, and b) aligning code fragments with corresponding comments. However, neither of these components is similar to the contrastive learning approach in LLaMoCo. As described in Section 3.2 (lines 262–269), LLaMoCo addresses a novel language alignment task with several unique challenges specific to the optimization field, including a) different prompts (optimization problems) often share similar solutions (optimizers), and b) similar prompts require different solutions. To address them, our proposed contrastive learning warmup aligns the hidden vectors in the final self-attention block of decoder-only LLMs, rather than aligning hidden dropout masks as in Unixcoder. By our efficient contrastive warmup (only 5 epochs), the learning effectiveness of the subsequent SFT process is significantly improved as shown in Figure 3.
> 3. We argue that the tasks in [3], which focus on daily conversations (a relatively simple domain), are significantly less complex compared to the optimization tasks addressed in LLaMoCo, which are challenging even for human experts. In optimization tasks, constructing well-defined problem descriptions is significantly more challenging than handling daily conversations. To address this, we introduced a templated problem construction process and designed tailored prompt descriptions. Moreover, labelling optimization problems requires expert-level knowledge to identify and fine-tune well-performing optimizers and to write the corresponding code. To ensure robustness, we conducted large-scale benchmarking and grid searches to determine competitive hyperparameters. Additionally, the unique cross-modal challenges described above necessitate an effective learning paradigm, which motivated the development of our contrastive learning warmup method.
>
> We will refine the introduction and methodology sections to highlight these discussions. We thank the reviewer for the valuable suggestions that strengthen our paper and enhance the discussion.
>
> **[W2, refining figures and writing]**
>
> We have addressed the typos you mentioned in the revised paper. Regarding Figure 1, we have updated it in the revised paper, where we illustrate all sub-components of the instruction tuning process including the dataset construction, benchmarking, and the two-phase fine-tuning in the figure to show the technical novelty of LLaMoCo compared with existing works.

---

> ### Author Response · Authors · 2024-11-22
> **Response to Reviewer #1QNj (part 2/2)**
>
> **[Q1, impact of dataset size]**
>
> We agree with the reviewer that an in-depth analysis of LLaMoCo’s scaling law would significantly enhance the impact of this work, considering both dataset size and model size. To this end, we selected the CodeGen family (350M, 1B, 3B, 7B) as the backbone model for LLaMoCo and conducted experiments with four training sets of varying sizes (1k, 5k, 15k, 30k). The optimization performance of these 16 trained models on the test set $\mathbb{I}_{eval}$ is presented in the following table:
>
>
>
> | model/data | 1k | 5k | 15k | 30k |
> | --- | --- | --- | --- | --- |
> | 350M | 47.260% | 66.661% | 80.306% | 81.843% |
> | 1B | 46.799% | 67.829% | 81.783% | 82.541% |
> | 3B | 47.131% | 68.492% | 82.501% | 83.315% |
> | 7B | 45.645% | 70.147% | 82.966% | **83.513%** |
>
> The results above yield several key observations: a) When the dataset size is very small (1k), increasing the model size does not result in performance gains, likely due to overfitting. b) For all model sizes, increasing the dataset size consistently improves performance. c) In summary, both model size and dataset size play crucial roles in determining the final performance of LLaMoCo. We have added the above results and discussion in lines 380-382 and Appendix E.3 of the revised paper (colorred in blue).
>
> **[Q2, data quality control]**
>
> The quality of the dataset is ensured by including only solvable and non-trivial problems. Specifically, for unconstrained problems, the composition and hybrid construction of base functions follow the procedure outlined in the IEEE CEC 2021 Single-Objective Competition, where the optimum and the optimal objective value of the constructed function are analytically derived. By applying rotation and shifting to the optimum, we modify the optimization landscape, ensuring the solution remains non-trivial. For constrained problems, we further validate solvability by running specialized optimizers from our algorithm pool on the constructed problem instances. These optimizers are executed multiple times (50 runs) to confirm the absence of constraint conflicts, ensuring that each problem instance is solvable. We have included this discussion in the Appendix A.3 of the revised paper (colorred in blue).
>
> **[Q3, computational overhead]**
>
> The increased token consumption by LLaMoCo can be attributed to two key factors: a) The baseline models we compare against tend to output very simple optimizers and sometimes incomplete codes due to their limited optimization knowledge. In contrast, LLaMoCo often generates more complex optimizer codes to achieve superior optimization performance. b) LLaMoCo includes user-friendly comments above each line of the generated code to help users understand and customize the content as needed, enhancing its flexibility. We kindly invite the reviewer to refer to Figures 7, 8, and 9 in Appendix F for a detailed examination of these two aspects. Nevertheless, as shown in Table 1, the increased token consumption in LLaMoCo is much less than the existing prompt for solution works such as OPRO (2.4k v.s. 115k) and prompt for optimizer works such as GPT-4 Turbo (2.4k v.s. 3.5k), which underscores the effectiveness and efficiency of LLaMoCo.
>
> We hope the above responses could enhance your confidence in our work.

---

> ### Author Response · Authors · 2024-11-28
> **Request for further feedback**
>
> Dear reviewer #1QNj:
>
> Since the discussion period has been extended, we respectifully request your feedback on our responses. In these responses, we have conducted additional experiments and provided in-depth discussions to address your concerns. If there are any further concerns, we are eager to continue this discussion and address them. We look forward to hearing from you.
>
> Best regards,
> the authors

---

> > ### Author Response · Authors · 2024-12-02
> >
> > Dear Reviewer #1QNj:
> >
> > The discussion period will end soon. We have provided point-to-point responses for your review comments. If you still have any concerns, we look forward to hearing from you and will address them before the discussion ends.
> >
> > Best regards, the authors

---

> > > ### Comment · Reviewer_1QNj · 2024-12-03
> > >
> > > I appreciate the authors for their responses!
> > >
> > > I have read the responses. I agree with the authors for their novelty and I'm satisfied with the experiment on data scaling.
> > >
> > > However, my knowledge in the optimization domain is limited and I cannot be sure how significant this work's contribution is to the optimization community. Thus, I decided to keep my score at 6 for acceptance in the field of code generation.

---

> > > > ### Author Response · Authors · 2024-12-03
> > > >
> > > > We appreciate the reviewer for acknowleding our work's novelty and contributions. Thanks for your precious time, valuable comments and positive feedback!

---

### Official Review · Reviewer_mqqR · 2024-11-04

**Soundness:** 3
**Presentation:** 3
**Contribution:** 2
**Rating:** 6
**Confidence:** 3

**Summary:**

This paper introduces LLaMoCo, a framework for fine-tuning general-purpose Large Language Models (LLMs) to generate optimization code through instruction tuning. The authors construct a specialized code-to-code instruction dataset tailored for optimization tasks. They enhance the training process with techniques such as contrastive warm-up, data augmentation via rephrasing, and balanced sampling. These methods are evaluated across three pre-trained models of different sizes (S, M, L), showing significant performance improvements. An ablation study further validates the effectiveness of the proposed techniques. Overall, the paper presents a promising approach to adapting LLMs for the specialized task of optimization code generation.

**Strengths:**

1. Specialized Dataset Creation: The development of a tailored code-to-code instruction dataset is a significant contribution. It aligns the fine-tuning process closely with the target task and provides a valuable resource for future research in optimization code generation.
2. Innovative Training Enhancements: Implementing contrastive warm-up, data augmentation through rephrasing, and balanced sampling demonstrates a comprehensive strategy to improve model performance. These techniques address common challenges in model training, such as overfitting and data imbalance.
3. Comprehensive Evaluation and Analysis: Evaluating the framework across models of varying sizes offers insights into scalability and the impact of model complexity. The inclusion of an ablation study allows for a deeper understanding of how each training enhancement contributes to the overall performance.

**Weaknesses:**

1. Unexpected Performance Across Model Sizes: Table 1, 2 and 3 show that the performance of LLaMoCo-S, LLaMoCo-M and LLaMoCo-L are very similar. The results also show that LLaMoCo-S sometimes outperforms its larger counterparts (LLaMoCo-M and LLaMoCo-L), despite having significantly fewer parameters. This is counterintuitive and raises concerns about potential inefficiencies in leveraging larger model’s increased capacity.

**Questions:**

1. Investigate Model Performance Discrepancies: It would be beneficial to analyze why the smaller model occasionally outperforms larger ones. This could involve examining the training dynamics, learning rates, or potential overfitting issues in larger models. Providing insights or adjustments based on this analysis would strengthen the validity of the results.
2. Expand Baseline Comparisons: Could the authors add another baseline of ChatGPT o1-mini/o1-preview? Since o1-mini/o1-preview are reasoning/coding/math enhanced models. I expect it to perform better than ChatGPT 4o. These models are designed for coding tasks and would serve as competitive benchmarks to better evaluate LLaMoCo's performance. Incorporating such comparisons would contextualize LLaMoCo's performance within the broader landscape of code generation research.
3. Enhance Robustness Evaluation: Assessing the models on out-of-distribution samples or real-world optimization problems beyond the dataset used for training could demonstrate the generalization capabilities and practical applicability of LLaMoCo, which could alleviate/address the concern of “Unexpected Performance Across Model Sizes”.

---

> ### Author Response · Authors · 2024-11-22
> **Response to Reviewer #mqqR (part 1/2)**
>
> We appreciate the reviewer for recognizing our LLaMoCo as a significant contribution to adapting LLMs for optimization code generation, supported by our dataset as a valuable resource for future work, an innovative training methodology, and a comprehensive experimental analysis. We hope the following responses address the remaining concerns effectively.
>
> **[W1 & Q1, model size & performance discrepancy]**
>
> We appreciate the observation! Firstly, we would like to clarify that larger models generally outperform smaller ones across Tables 1 to 3, indicating no overfitting under the given data scale. We acknowledge one exception: in Table 1, the medium model (LLaMoCo-M) outperforms the large model (LLaMoCo-L) on unconstrained optimization problems. This discrepancy likely stems from differences in the base models used during fine-tuning. Specifically, LLaMoCo-S, LLaMoCo-M, and LLaMoCo-L are fine-tuned on CodeGen-Mono (350M), Phi-2 (2.7B), and Code Llama (7B), respectively, demonstrating the generalizability of LLaMoCo across various backbone models. These base model differences may contribute to the observed performance variation.
>
> Notably, we actually have another table used to validate the scaling law of our LLaMoCo. Specifically, the experiment presented in Table 2 uses the CodeGen-Mono model series (ranging from 350M to 7B) serving as the base for LLaMoCo. The results indeed demonstrate that performance improves as model size increases, confirming that larger models achieve significant performance gains under our data scale. We present this table below for your convenience:
>
> | Model Size | 350M | 1B | 3B | 7B |
> | --- | --- | --- | --- | --- |
> | CodeGen-Mono  | 15.341% | 18.943% | 19.348% | 20.982% |
> | LLaMoCo-CodeGen | **81.843%** | **82.541%** | **83.315%** | **83.513%** |
>
> Moreover, we would like to clarify that while the results in the table may appear similar, they are based on a **normalized performance metric** (detailed in Appendix D), which scales optimization performance by the objective value range of the target problem instance. These ranges can be extremely large (e.g., $10^{20}$), causing performance differences to appear smaller than they are. For instance, in Table 1, the small model (LLaMoCo-S) achieves a performance of 81.843%, which seems "similar" to the medium model (LLaMoCo-M) at 83.369%. However, this 1.5% difference reflects a significant performance gap in absolute terms. The reason we normalize the optimization performance is that given various objective value scales in different problem instances, it is unreasonable to compute the absolute average final objective value across problem instances. Additionally, we believe that such normalization improves the table readability of the results.
>
> **[Q2, add more baselines]**
>
> Thank you for the suggestion! Following your suggestion, we have tested three additional OpenAI models: **GPT-4o, o1-mini and o1-preview**, on our test dataset $\mathbb{I}_{eval}$.  Below, we provide a table comparing their performance with the GPT-4 Turbo baseline and our LLaMoCo mdoels:
>
> | Metric | GPT-4 Turbo | GPT-4o | o1-mini | o1-preview | LLaMoCo-S | LLaMoCo-M | LLaMoCo-L |
> | --- | --- | --- | --- | --- | --- | --- | --- |
> | Err. | 41.667% | 33.771% | **3.355%** | 4.107% | 5.580% | 5.434% | 5.509% |
> | Rec. | 13.072% | 14.405% | **10.299%** | 10.641% | 10.826% | 10.349% | 10.461% |
> | Perf. | 74.248% | 75.193% | 80.269% | 79.945% | 81.843% | 83.369% | **83.451%** |
> | Comp. | 3.5k | 3.6k | 4.1k | 4.1k | 2.4k | 2.4k | **2.4k** |
>
> From the results, we observe the following:
>
> 1. o1-mini/preview v.s. GPT-4o: The o1 models achieve significantly lower coding errors compared to the GPT-4o model, demonstrating their robust coding enhancement capabilities.
> 2.  o1-mini v.s. LLaMoCo: On one hand, the error rate of o1-mini is lower than that of our LLaMoCo, primarily due to o1-mini's black-box training on an extremely large coding task. On the other hand, our LLaMoCo, despite being trained on a much smaller model, achieves greater optimization performance gains while consuming fewer tokens. Furthermore, we analyzed the source codes generated by o1-mini, as we did for the GPT-4 Turbo model in Section 4.4. It was found that o1-mini also tends to generate a specific optimizer, the DE algorithm, for nearly all tested problems. This observation reinforces the core motivation behind LLaMoCo, which is to explore how domain-specific knowledge can be effectively injected into large language models to adapt them for specialized scientific tasks.
>
> We have updated the above discussion into our revised paper, Section 4.4 (colorred in blue). We kindly request the reviewer to examine it.

---

> ### Author Response · Authors · 2024-11-22
> **Response to Reviewer #mqqR (part 2/2)**
>
> **[Q3, out-of-distribution evaluation]**
>
> We would like to clarify that the **out-of-distribution evaluation has already been conducted and is presented in Table 3**. The tested problems in this table come from diverse realistic scenarios with significantly different problem structures compared with the synthetic problems we used for training. The results demonstrate two key points:
>
> 1. The LLMs fine-tuned by LLaMoCo exhibit superior generalization performance to other baselines on the out-of-distribution tasks.
> 2. The incremental performance trend among the three LLaMoCo models (S, M, L) in Table 3 is consistent with the trend in the in-distribution evaluation in Table 1.

---

> ### Author Response · Authors · 2024-11-26
> **Request for further feedback**
>
> Dear reviewer #mqqR, since the discussion period is extended, we respectifully request you to check the experimental restults and discussion we have added following your constructive suggestions. We look forward to your futher feedback to help us improve this paper. We are open to any suggestions from you. Thanks for your precious time!

---

> > ### Comment · Reviewer_mqqR · 2024-11-27
> >
> > I appreciate the authors' efforts in addressing my questions. As most of my concerns have been satisfactorily addressed, I will update my score and recommend your paper for acceptance. Thank you for your detailed responses and clarifications!

---

> > > ### Author Response · Authors · 2024-11-27
> > >
> > > We sincerely appreciate your positive feedback on our LLaMoCo! Thanks for the time and efforts you have contributed to improve our paper.

---

### Author Response · Authors · 2024-12-04
**Global Response**

We would like to express our sincere gratitude for the time and effort the reviewers and AC have invested in reviewing our paper. First of all , we are so honored that LLaMoCo has been recognized as a **significant contribution to optimization community** (Reviewer #mqqR, #1QNj and #eFRN) and **code generation community** (Reviewer #mqqR, #1QNj). We are also pleased to see the reviewers have commended LLaMoCo for its **novel framework** (Reviewer #mqqR, #1QNj and #eFRN), **valuable dataset proposal** (all reviewers), **innovative training paradigm** (Reviewer #mqqR, #1QNj and #eFRN), **comprehensive evaluation** (Reviewer #mqqR and #eFRN) and **superior optimization performance** (all reviewers).

In this global response, we primarily summarize common suggestions shared by the reviewers and provide an overview of the additional discussion with experimental results that address these suggestions, as follows.

---

**[Adding more baselines for comparison, Reviewer #mqqR, #ugM1 and #eFRN]**

1. Reviewer #mqqR suggests adding GPT-4o, o1-mini and o1-preview for a comprehensive comparison, which we have included them in **Section 4.4 and Table 4** (highlighted in blue) of the revised paper.
2. Reviewer #ugM1 suggets adding GPT-4 Vector Search as an in-context learning baseline, which we have also included them in **Table 4** of the revised paper. These results further validate the superiority of LLaMoCo.
3. Reviewer #eFRN suggests adding some automatic algorithm selection methods as baselines. We have clarified the fundamental differences between LLaMoCo and algorithm selection methods, noting that direct comparison would be unfair. This explanation has been added in **Section 4.1, lines 349-354** (highlighted in blue) of the revised paper.

**[Exploring LLaMoCo’s scalling law, Reviewer #mqqR and #1QNj]**

Reviewer #mqqR and #iQNj suggest exploring the impact of the model capacity and dataset size on LLaMoCo’s performance respectively. We hence provide an additional scaling law experiment on LLaMoCo in **Section 4.2, Table 2** and **Appendix E.3, Table 9**, where we find that both model capacity and dataset size play key roles in the final performance and the setting we adopt in the paper achieves best results.

**[Evaluating LLaMoCo on real-world scenarios, Reviewer #mqqR and #ugM1]**

1. Reviewer #mqqR requests the performance of LLaMoCo on realistic problems. We have clarified that results for eight real-world problems from diverse domains are already included in **Section 4.2, Table 3** of the original paper.
2. Reviewer #ugM1 suggests adding more realistic problem instances to further validate the generalization ability of LLaMoCo. Following the suggestion, we have included a comprehensive real-world problem collection of 57 engineering problems and update the results in **Appendix E.2, Table 8** (highlighted in blue) of the revised paper, which provide a clear evidence of LLaMoCo’s superior generalization performance.

**[Highlighting novelties of LLaMoCo, Reviewer #1QNj and #eFRN]**

Following the reviewers’ suggestions, we have discussed the challenges of fine-tuning general LLMs for optimization domain and the corresponding novelties of LLaMoCo to address these challenges. We have added this dicussion into our revised paper (Section 2, lines 125-128, 142-145, 169-173; Section 3, 181-187; all highlighted in blue) to highlight our novel methodology.

**[Validating the ablation robustness, Reviewer #ugM1 and #eFRN]**

Reviewer #ugM1 and #eFRN suggest adding the ablation results on the proposed contrastive warm-up strategy on all combinations of the foundation LLMs and training problem sets. Following the suggestion, we have added these results in **Appendix E.5, Figure 6** of the revised paper, where the contrastive warm-up strategy consistently improves the instruction-tuning of LLaMoCo under various settings.

---

We hope the above summary of our discussion with the reviewers could provide convenience for reviewers and AC to grasp all focused issues and all efforts we have made to address them.

---

### Meta-Review · Area_Chair_Ksno · 2024-12-05

**Metareview:**

### Summary of Claims and Findings
The paper introduces LLaMoCo, a novel instruction-tuning framework for adapting large language models (LLMs) to generate optimization code directly from problem descriptions in Python or LaTeX. The framework features a curated dataset of optimization problems paired with expert-level solutions and a two-phase training strategy that includes a contrastive warm-up phase to align problem-solution representations. Experimental results demonstrate that LLaMoCo significantly outperforms GPT-4 Turbo and other baselines in optimization performance across synthetic and realistic tasks.

### Strengths
1. **Framework Innovation**: LLaMoCo offers a structured approach to instruction tuning in the optimization domain, addressing challenges such as problem representation, model alignment, and data imbalance.
2. **Comprehensive Evaluation**: The paper demonstrates strong results on a diverse set of synthetic and real-world problems, coupled with scaling law analysis and ablation studies.
3. **Practical Impact**: The model's ability to generate efficient and effective optimization code has broad applications in engineering and optimization research.

### Weaknesses
1. **Perceived Incrementality**: Reviewers noted that many components (e.g., instruction tuning, dataset construction, contrastive learning) leverage existing techniques. Despite the authors’ rebuttal, the domain-specific nature of the contributions remains under-emphasized.
2. **Limited Demonstration of Contrastive Warm-Up's Necessity**: While the warm-up strategy appears effective, its justification as domain-specific was unconvincing to some reviewers, with suggestions that it could apply broadly across domains.
3. **Modest Scaling Gains**: Performance improvements from smaller to larger models appeared incremental, raising questions about the optimization knowledge effectively learned.

### Decision
While LLaMoCo represents a commendable effort to adapt LLMs for optimization code generation, the core novelty remains a concern. The primary contributions are seen as applications of existing methods rather than groundbreaking domain-specific innovations. Combined with lingering skepticism about the necessity and specialization of the proposed techniques, this submission falls marginally below the acceptance threshold.

**Additional Comments On Reviewer Discussion:**

- **Novelty and Domain-Specificity**: Some reviewers revised their ratings upward, appreciating the authors’ explanations and additional experiments highlighting the dataset's construction and the training strategy's uniqueness. However, one reviewer remained unconvinced about the "unique technical challenges" of the optimization domain.
- **Experimental Updates**: Authors added new baselines (e.g., GPT-4 with vector search), zero-shot evaluations on 57 real-world problems, and scaling law experiments. These were positively received, with reviewers noting the effort and thoroughness.
- **Contrastive Warm-Up**: Despite its positive impact on performance, one reviewer suggested separating it into a standalone paper due to its potential generalizability.

---

### Decision · Program_Chairs · 2025-01-22

Reject